# Flexible Infinite-Width Graph Convolutional Neural Networks

**Ben Anson**                                           *ben.anson@bristol.ac.uk*
*School of Mathematics*
*University of Bristol*
*Bristol, United Kingdom*

**Edward Milsom**                                       *edward.milsom@bristol.ac.uk*
*School of Mathematics*
*University of Bristol*
*Bristol, United Kingdom*

**Laurence Aitchison**                                  *laurence.aitchison@bristol.ac.uk*
*School of Computer Science*
*University of Bristol*
*Bristol, United Kingdom*

**Reviewed on OpenReview:** *https://openreview.net/forum?id=Q2M4yijKSo*

## Abstract

A common theoretical approach to understanding neural networks is to take an infinite-width limit, at which point the outputs become Gaussian process (GP) distributed. This is known as a neural network Gaussian process (NNGP). However, the NNGP kernel is fixed and tunable only through a small number of hyperparameters, thus eliminating the possibility of representation learning. This contrasts with finite-width NNs, which are often believed to perform well because they are able to flexibly learn representations for the task at hand. Thus, in simplifying NNs to make them theoretically tractable, NNGPs may eliminate precisely what makes them work well (representation learning). This motivated us to understand whether representation learning is necessary in a range of graph tasks. We develop a precise tool for this task, the graph convolutional deep kernel machine. This is very similar to an NNGP, in that it is an infinite width limit and uses kernels, but comes with a "knob" to control the amount of flexibility and hence representation learning. We found that representation learning gives noticeable performance improvements for heterophilous node classification tasks, but less so for homophilous node classification tasks.

## 1 Introduction

A fundamental theoretical method for analyzing neural networks involves taking an infinite-width limit of a randomly initialized network. In this setting, the outputs converge to a Gaussian process (GP), and this GP is known as a neural network Gaussian process or NNGP (Neal, 1996; Lee et al., 2018; Matthews et al., 2018). NNGPs have been used to study various different neural network architectures, ranging from fully-connected (Lee et al., 2018) to convolutional (Novak et al., 2018; Garriga-Alonso et al., 2018) and graph networks (Walker & Glocker, 2019; Niu et al., 2023).

However, it is important to understand whether these NNGP results apply in practical neural network settings. One way of assessing the applicability of NNGP results is to look at performance. Specifically,

if infinite-width NNGPs perform comparably with finite-width NNs, we can reasonably claim that infinite-width NNGPs capture the "essence" of what makes finite-width NNs work so well. In contrast, if finite-width NNs perform better, that would indicate that NNGPs are missing something.

Why might NNGPs underperform relative to NNs? One key property of the infinite-width NNGP limit is that it eliminates representation learning: the kernel of the GP is fixed, and cannot be tuned except through a very small number of hyperparameters. This fixed kernel makes it straightforward to analyze the behaviour of networks in the NNGP limit. However, the elegant fixed kernel comes at a cost. In particular, the top-layer representation/kernel is highly flexible in finite networks, and this flexibility is critical to the excellent performance of deep networks (Bengio et al., 2013; LeCun et al., 2015). This indicates that in some cases, the infinite-width NNGP limit can "throw the baby out with the bathwater": in trying to simplify the system to enable theoretical analysis, the NNGP limit can eliminate some of the most important properties of deep networks that lead to them performing well.

In the setting of convolutional networks for CIFAR-10, this is precisely what was found: namely, that infinite-width NNGPs underperform finite-width NNs (Adlam et al., 2023; Garriga-Alonso et al., 2018; Shankar et al., 2020), and it has been hypothesised that this difference arises due to the lack of representation learning in NNGP (Aitchison, 2020; MacKay et al., 1998).

In this paper, we consider the question of NNGP performance in the Graph Convolutional Network (GCN) setting. Surprisingly, prior work has shown that the graph convolutional NNGP performs comparably to the GCN (Niu et al., 2023), perhaps suggesting that representation learning is not important in graph settings.

However, our work indicates that this picture is incomplete. In particular, we show that while the graph convolutional NNGP is competitive on some datasets, it is much worse than GCNs on other datasets. This would suggest a hypothesis that representation learning in graph tasks is dataset dependent, with some datasets needing representation learning, and others not.

Importantly, testing this hypothesis rigorously is difficult, as there are many differences between the infinite-width, fixed representation graph convolutional NNGP and the finite-width, flexible representation GCN, not just that the GCN has representation learning. Perhaps the best approach to testing the hypothesis would be to develop a variant of the NNGP with representation learning, and to see how performance changed as we altered the amount of representation learning allowed by this new model. However, this is not possible with the traditional NNGP framework. Instead, we need to turn to recently developed deep kernel machines (DKMs) (Yang et al., 2023; Milsom et al., 2023). DKMs, like NNGPs, are obtained via an infinite-width limit, and in practice work entirely in terms of kernels. The key difference is that DKMs allow for representation learning, while NNGPs do not. Specifically, DKMs have a tunable parameter, $\nu$, which controls the degree of representation learning. For small values of $\nu$, DKM representations are highly flexible and learned from data. In contrast, as $\nu \to \infty$, flexibility is eliminated, and the DKM becomes equivalent to the NNGP. The graph convolutional DKM therefore forms a precise tool for studying the need for representation learning in graph tasks.

Using the graph convolutional DKM, we examined the need for representation learning in a variety of graph tasks. Node classification tasks in particular exist on a spectrum from homophilous, where adjacent nodes in a graph tend to have similar labels, to heterophilous, where adjacent nodes tend to be dissimilar. We found that homophilous tasks tended not to require representation learning, while heterophilous tasks did require representation learning.

Concretely, our contributions are as follows:

- We develop a graph convolutional variant of deep kernel machines (Section 4).

- We propose two scalable inducing-point approximation schemes for the graph convolutional DKM (Section 4.2, Appendix B).

- We analyze the graph convolutional DKM in the linear setting, and provide a closed-form solution for the learned representations (Section 5).

- By considering the performance of graph convolutional NNGPs relative to GCNs and by tuning $\nu$ in the graph convolutional DKM, we find that representation learning improves performance in heterophilous node classification tasks but not homophilous node classification tasks.

## 2 Related Work

Graph convolutional networks are a type of graph neural network (Scarselli et al., 2008; Kipf & Welling, 2017; Bronstein et al., 2017; Velicković et al., 2017) originally introduced by Kipf & Welling (2017), motivated by a localised first-order approximation of spectral graph convolutions. They use the adjacency matrix to aggregate information from a node's neighbours at each layer of the neural network. When first published, graph convolutional networks outperformed existing approaches for semi-supervised node classification by a significant margin.

Homophily, a measure of similarity between connected nodes in a graph, has been studied extensively in the context of GNNs (Luan et al., 2024; Zhu et al., 2020; 2021; Maurya et al., 2021; Platonov et al., 2023), mostly to point out that lack of homophily (meaning that connected nodes tend to be dissimilar) is detrimental to performance in GNNs. Here, we examine an adjacent but different matter: the relationship between homophily and representation learning.

Infinite-width neural networks were first considered in the 1990s (Neal, 1996; Williams, 1996) where they only considered shallow networks at initialisation. Cho & Saul (2009) derived an equivalence between deep ReLU networks and arccosine kernels. This was later generalised to arbitrary activation functions by Lee et al. (2018), who used a Bayesian framework to show that any infinitely-wide deep neural network is equivalent to a Gaussian process, which they dubbed the neural network Gaussian process (NNGP) (Lee et al., 2020; 2018; Matthews et al., 2018). The NNGP has since been extended, for example to accommodate convolutions (Garriga-Alonso et al., 2018; Novak et al., 2018), graph structure (Niu et al., 2023; Hu et al., 2020a) and more (Yang, 2019).

Using a similar infinite-width limit, but on SGD-trained networks rather than Bayesian networks, Jacot et al. (2018) studied the dynamics of neural network training under gradient descent, deriving the neural tangent kernel (NTK), which describes the evolution of the infinite-width network over time. The NTK suffers from an analogous problem to the NNGP's lack of representation learning, since the NTK limit implies that the intermediate features do not evolve over time (Yang & Hu, 2021; Bordelon & Pehlevan, 2023; Vyas et al., 2023). Alternative limits, such as the recent $\mu$-P parameterisation (Yang & Hu, 2021) fix this problem by altering the scaling of parameters. However, the $\mu$-P line of work only tells us that feature/representation learning will happen, not what the learned features/representations will be. In contrast the DKM framework gives us the exact learned representations.

Applying *fixed* kernel functions to graphs is a well explored endeavour (Shervashidze & Borgwardt, 2009; Kashima et al., 2003; Shervashidze et al., 2009). Kernels can be defined directly for graphs and applied in a shallow fashion, shallow kernel can be stacked (Achten et al., 2023), or kernels can be used as a component of a larger deep learning algorithm (Cosmo et al., 2021; Yanardag & Vishwanathan, 2015). Our work differs fundamentally in that kernels themselves are learned, rather than learning features and then applying a fixed kernel.

## 3 Background

We give a background on deep Gaussian processes (DGPs), and how under an infinite-width limit they become neural network Gaussian processes (NNGPs). We review how modifying this infinite-width limit gives flexible DKMs, in contrast to NNGPs which have fixed kernels. We also give an overview of the graph convolutional network (GCN) and the graph convolutional NNGP. These ingredients allow us to define a DKM in the graph domain, a so-called "graph convolutional DKM", in Section 4.

### 3.1 Neural Networks as Deep Gaussian Processes

Consider a fully-connected NN with inputs $\mathbf{X} \in \mathbb{R}^{P \times \nu_0}$, outputs/labels $\mathbf{Y} \in \mathbb{R}^{P \times \nu_{L+1}}$, and intermediate-layer features $\mathbf{F}^\ell \in \mathbb{R}^{P \times N_\ell}$. By $P$ we refer to the number of datapoints, $\nu_0$ and $\nu_{L+1}$ the number of input and output features respectively. The width of each layer is denoted $N_\ell = N\nu_\ell$, where $\nu_\ell$ is fixed, and $N$ allows us to scale the width of all layers simultaneously. We compute features at each layer using the previous layer via $\mathbf{F}^\ell = \phi(\mathbf{F}^{\ell-1})\mathbf{W}^\ell$, where $\mathbf{W}^\ell \in \mathbb{R}^{N_{\ell-1} \times N_\ell}$ are the weights and $\phi$ is the pointwise nonlinearity (e.g. ReLU). With a Gaussian prior on the weights, $W_{ij}^\ell \sim \mathcal{N}(0, \frac{1}{N_{\ell-1}})$, the conditional distribution of features is given by,

$$P(\mathbf{F}^\ell \mid \mathbf{F}^{\ell-1}) = \prod_{\lambda=1}^{N_\ell} \mathcal{N}(\mathbf{f}_\lambda^\ell; \mathbf{0}, \mathbf{K}_{\text{features}}(\mathbf{F}^{\ell-1})), \tag{1a}$$

$$P(\mathbf{Y} \mid \mathbf{F}^L) = \prod_{\lambda=1}^{\nu_{L+1}} \mathcal{N}(\mathbf{y}_\lambda; \mathbf{0}, \mathbf{K}_{\text{features}}(\mathbf{F}^L) + \sigma^2\mathbf{I}), \tag{1b}$$

where $\mathbf{K}_{\text{features}}(\mathbf{X}) = \frac{1}{N_\ell}\phi(\mathbf{X})\phi(\mathbf{X})^T$, and the subscript $\lambda$ refers to the $\lambda$th feature/column, so that $\mathbf{F}^\ell = (\mathbf{f}_1^\ell \ \mathbf{f}_2^\ell \ \cdots \ \mathbf{f}_{N_\ell}^\ell)$. Eq. (1) is equivalent to a Bayesian NN, but in the above form it is usually called a deep Gaussian process (DGP) (Damianou & Lawrence, 2013; Salimbeni & Deisenroth, 2017), albeit with an unusual kernel. We can generalize Eq. (1) to allow alternative kernels $\mathbf{K}_{\text{features}}(\cdot)$ such as the squared exponential kernel.

### 3.2 Deep Gaussian Processes in Terms of Gram Matrices

For certain kernel functions (e.g. arccosine kernels (Cho & Saul, 2009) and RBF), features are unnecessary, and inter-layer dependencies in a DGP can be summarised entirely by Gram matrices $\mathbf{G}^\ell = \frac{1}{N_\ell}\mathbf{F}^\ell(\mathbf{F}^\ell)^T \in \mathbb{R}^{P \times P}$. For these kernels, we are able to compute the kernel from the Gram matrix alone, and thus there is a function over Gram matrices $\mathbf{K}(\cdot) : \mathbb{R}^{P \times P} \to \mathbb{R}^{P \times P}$ such that $\mathbf{K}_{\text{features}}(\mathbf{F}^\ell) = \mathbf{K}(\mathbf{G}^\ell)$ [1]. We can therefore rewrite the DGP in Eq. (1) as,

$$P(\mathbf{F}^\ell \mid \mathbf{G}^{\ell-1}) = \prod_{\lambda=1}^{N_\ell} \mathcal{N}(\mathbf{f}_\lambda^\ell; \mathbf{0}, \mathbf{K}(\mathbf{G}^{\ell-1})), \tag{2a}$$

$$P(\mathbf{Y} \mid \mathbf{G}^L) = \prod_{\lambda=1}^{\nu_{L+1}} \mathcal{N}(\mathbf{y}_\lambda; \mathbf{0}, \mathbf{K}(\mathbf{G}^L) + \sigma^2\mathbf{I}), \tag{2b}$$

where all inter-layer dependencies in Eq. (2) are expressed using only Gram matrices. We can remove features altogether, and reframe the model purely in terms of Gram matrices (Aitchison et al., 2021),

$$P(\mathbf{G}^\ell \mid \mathbf{G}^{\ell-1}) = \mathcal{W}(\mathbf{G}^\ell; \mathbf{G}^{\ell-1}/N_\ell, N_\ell), \tag{3a}$$

$$P(\mathbf{Y} \mid \mathbf{G}^L) = \prod_{\lambda=1}^{\nu_{L+1}} \mathcal{N}(\mathbf{y}_\lambda; \mathbf{0}, \mathbf{K}(\mathbf{G}^L) + \sigma^2\mathbf{I}), \tag{3b}$$

where $\mathcal{W}(\mathbf{V}, n)$ is the Wishart distribution with scale matrix $\mathbf{V}$ and degrees of freedom $n$ (Gupta & Nagar, 2018).

### 3.3 Wide Deep Gaussian Processes

Consider a deep ReLU network. Bayesian inference over weights/features is not tractable, but remarkably, there *is* a tractable closed form solution if we take an infinite width limit. To see this, first observe that we

---

[1] For avoidance of doubt, $\mathbf{K}_{\text{features}}(\cdot) : \mathbb{R}^{P \times N_\ell} \to \mathbb{R}^{P \times P}$ computes the kernel from the features, while $\mathbf{K}(\cdot) : \mathbb{R}^{P \times P} \to \mathbb{R}^{P \times P}$ equivalently computes the same kernel from the corresponding Gram matrix.

can calculate the kernel non-linearity without features; Cho & Saul (2009) showed that in the limit $N_\ell \to \infty$, ReLU kernels are equivalent to arccosine kernels:

$$\phi = \text{ReLU} \implies \lim_{N_\ell \to \infty} \tfrac{1}{N_\ell} \phi(\mathbf{F}^\ell)\phi^T(\mathbf{F}^\ell) = \mathbf{K}_{\text{arccos}}(\mathbf{G}^\ell). \tag{4}$$

Further, the arccosine kernel is easily computable,

$$K_{\text{arccos}}(\mathbf{G})_{ij} = \frac{\sqrt{G_{ii}G_{jj}}}{\pi}\left(\sin\theta_{ij} + (\pi - \theta_{ij})\cos\theta_{ij}\right), \text{ where } \cos\theta_{ij} = \frac{G_{ij}}{\sqrt{G_{ii}G_{jj}}}. \tag{5}$$

Secondly, Gram matrices become deterministic in wide networks, since they can be understood as an average of IID features, $\mathbf{G}^\ell = \frac{1}{N_\ell}\sum_{\lambda=1}^{N_\ell}\mathbf{f}_\lambda^\ell(\mathbf{f}_\lambda^\ell)^T$. By Eq. (1) or Eq. (2), the expectation of $\mathbf{G}^\ell$ conditioned on the previous layer is, $\mathbb{E}[\mathbf{G}^\ell|\mathbf{G}^{\ell-1}] = \mathbf{K}(\mathbf{G}^{\ell-1})$, while the variance scales with $1/N_\ell$. Thus, as we take an infinite-width limit, $N_\ell \to \infty$, the variance goes to zero, and the prior over the Gram matrix at layer $\ell$, $\mathbf{G}^\ell$, becomes deterministic and concentrated at its expectation, $\mathbf{K}(\mathbf{G}^{\ell-1})$. Thus, the kernel at every layer is a fixed deterministic function of the inputs that can be computed recursively,

$$\mathbf{K}_{\text{network}}(\mathbf{F}^\ell) = \underbrace{(\mathbf{K} \circ \cdots \circ \mathbf{K})}_{\ell \text{ times}}(\tfrac{1}{\nu_0}\mathbf{X}\mathbf{X}^T), \tag{6}$$

where $\frac{1}{\nu_0}\mathbf{X}\mathbf{X}^T$ is the kernel for the inputs, and $\mathbf{F}^0 = \mathbf{X}$. Setting $\ell = L$, we see that Eq. (6) holds true at the output layer, implying that the outputs of a NN or DGP are GP distributed with a fixed kernel in the infinite-width limit. This is known as the neural network Gaussian process (NNGP) (Lee et al., 2018).

### 3.4 Deep Kernel Machines

The clean analytic form of the NNGP kernel (Eq. 6) unfortunately has a big weakness: it underperforms finite-width counterparts empirically in many tasks (Aitchison, 2020; Pleiss & Cunningham, 2021). The NNGP kernel is a fixed function of inputs, meaning that no representation learning can occur, and the kernel cannot be shaped by the output labels. This is a big problem, since representation learning is widely understood to be central to the success of modern deep learning systems.

DKMs (Yang et al., 2023; Milsom et al., 2023) are a solution to the lack of representation learning in infinite-width NNGPs. DKMs can still be understood as an infinite-width limit of a DGP/NN, but the limit is altered to retain flexibility and allow learning of representations. In a DKM, Gram matrices are treated as learned parameters, rather than random variables as in a DGP or NNGP. Yang et al. (2023) derive the following "DKM objective" for fully-connected networks that can be optimized with respect to Gram matrices $\mathbf{G}^\ell$,

$$\mathcal{L}(\mathbf{G}^1, \ldots, \mathbf{G}^L) = \log \mathrm{P}(\mathbf{Y} \mid \mathbf{G}^L) - \sum_{\ell=1}^{L} \nu_\ell \mathrm{D}_{\text{KL}}\left(\mathcal{N}(\mathbf{0}, \mathbf{G}^\ell) \,\|\, \mathcal{N}(\mathbf{0}, \mathbf{K}(\mathbf{G}^{\ell-1}))\right). \tag{7}$$

Here, the likelihood term $\mathrm{P}(\mathbf{Y} \mid \mathbf{G}^L)$ measures how well the final layer representation $\mathbf{G}^L$ performs at the task. Under the derivation of Yang et al. (2023), $\mathrm{P}(\mathbf{Y} \mid \mathbf{G}^L)$ is a likelihood for a Gaussian process distribution over the outputs with kernel $\mathbf{K}(\mathbf{G}^L)$. The KL divergence terms are a measure of how much the Gram matrices $\mathbf{G}^1, \ldots, \mathbf{G}^L$ deviate from the NNGP. This can be seen due to the fact that a negative KL divergence is maximized exactly when the two distributions it is measuring are equal, i.e. $\mathbf{G}^\ell = \mathbf{K}(\mathbf{G}^{\ell-1})$. The DKM objective trades off the marginal likelihood against the KL divergence terms, with the KL terms acting as a regularizer towards the NNGP. The amount of regularization depends on the $\nu_\ell$ coefficients, and as we send $\nu_\ell \to \infty$, the DKM becomes equivalent to the standard infinite-width NNGP. As $\nu_\ell$ becomes smaller, we allow more flexibility in the Gram representations. Thus, through $\nu_\ell$, the DKM gives us a knob to tune the amount of representation learning allowed in our model.

### 3.5 Graph Convolutional Networks and their Wide counterparts

Kipf & Welling (2017) derived Graph Convolutional Networks (GCNs) by applying approximations to spectral methods on graphs. In a GCN layer, an adjacency matrix $\mathbf{A} \in P \times P$ is used to convolve (or 'mixup')

intermediate layer nodes. Mathematically, GCN post-activations are given by $\mathbf{H}^\ell = \phi(\hat{\mathbf{A}}\mathbf{H}^{\ell-1}\mathbf{W}^\ell)$, where $\hat{\mathbf{A}}$ is a normalized version of $\mathbf{A}$.

Similarly to Section 3.2, it is possible to construct an NNGP for the graph domain. The pre-activations $\mathbf{F}^\ell$ of a graph convolution layer are $\mathbf{F}^\ell = \hat{\mathbf{A}}\mathbf{H}^{\ell-1}\mathbf{W}^\ell$. Again, placing an IID Gaussian prior over our weights $W_{\mu\lambda}^\ell \sim \mathcal{N}(0, \frac{1}{N_{\ell-1}})$, the NNGP construction tells us that as we make the layers wide ($N_\ell \to \infty$), the pre-activations become GP distributed with kernel matrix $\mathbf{K} = \mathbb{E}[\mathbf{f}_\lambda^\ell(\mathbf{f}_\lambda^\ell)^T]$. We show in Appendix C that this expectation has closed-form,

$$\mathbf{K} = \hat{\mathbf{A}}\mathbf{\Phi}^{\ell-1}\hat{\mathbf{A}}^T, \tag{8}$$

where $\mathbf{\Phi}^{\ell-1} = \mathbb{E}[\mathbf{h}_\lambda^{\ell-1}(\mathbf{h}_\lambda^{\ell-1})^T]$ is the NNGP kernel of a fully connected network (e.g. the arccosine kernel in the case of a ReLU network). Using this kernel in the NNGP defined in Eq. 6 gives the graph convolutional NNGP recursion,

$$\mathbf{G}^\ell = \hat{\mathbf{A}}\mathbf{K}(\mathbf{G}^{\ell-1})\hat{\mathbf{A}}^T = \hat{\mathbf{A}}\mathbf{K}(\hat{\mathbf{A}}\mathbf{K}(\cdots\hat{\mathbf{A}}\mathbf{K}(\frac{1}{\nu_0}\mathbf{X}\mathbf{X}^T)\hat{\mathbf{A}}^T\cdots)\hat{\mathbf{A}}^T)\hat{\mathbf{A}}^T. \tag{9}$$

GCNs and graph convolutional NNGPs can be applied easily to node classification problems, since features or nodes are modelled directly. They can also be used for graph classification by applying mean pooling at the output layer.

In Sections 5 and 6 we consider homophily in graphs and its effect on graph convolutional NNGP performance. By homophily, we refer to edge homophily $h \in [0, 1]$, which is calculated as the proportion of edges $(j, k) \in \mathcal{E}$ that are between nodes with the same class.

## 4 Methods

In Section 3 we have seen how graph convolutional NNGP kernels can be calculated. We have also seen how DKMs add flexibility to fully-connected NNGPs. We now combine these two ideas to obtain a flexible infinite-width graph network — the 'graph convolutional DKM' — and develop an inducing point scheme to allow training on large datasets.

### 4.1 Graph Convolutional Deep Kernel Machines

The graph convolutional NNGP suffers from the "fixed representation" problem that plagues all NNGP models — its kernel is a fixed transformation of the inputs, regardless of the target labels. This means that there is little flexibility (apart from kernel hyperparameters) to learn suitable features for the task at hand. To solve this problem we develop the graph convolutional DKM, which has learnable kernel representations at each layer.

We arrive at a graph convolutional DKM by considering the Gram matrices in a DGP with graph mixups,

$$P(\mathbf{F}^\ell \mid \mathbf{F}^{\ell-1}) = \prod_{\lambda=1}^{N_\ell} \mathcal{N}(\mathbf{f}_\lambda^\ell; \mathbf{0}, \hat{\mathbf{A}}\mathbf{K}(\mathbf{G}^{\ell-1})\hat{\mathbf{A}}^T), \tag{10}$$
$$\text{where } \mathbf{G}^{\ell-1} = \frac{1}{N_{\ell-1}}\mathbf{F}^{\ell-1}(\mathbf{F}^{\ell-1})^T.$$

By taking the representation learning limit of the graph convolutional DGP, we show in Appendix A that the posterior over Gram matrices is point distributed; in other words, the Gram matrices are deterministic. Moreover, the Gram matrices maximize the graph DKM objective,

$$\mathcal{L}(\mathbf{G}^1, \ldots, \mathbf{G}^L) = \log P(\mathbf{Y} \mid \mathbf{G}^L) - \sum_{\ell=1}^{L} \nu_\ell D_{KL}\left(\mathcal{N}(\mathbf{0}, \mathbf{G}^\ell) \,\|\, \mathcal{N}(\mathbf{0}, \hat{\mathbf{A}}\mathbf{K}(\mathbf{G}^{\ell-1})\hat{\mathbf{A}}^T)\right). \tag{11}$$

The objective in Eq. (11) can be interpreted as a fully-connected DKM, but with a modified kernel. Again, just as in Section 3.4, we trade-off the likelihood with the KL divergence terms. That is, we trade-off

the suitability of the top-layer representation for explaining the labels $\mathbf{Y}$ versus divergence from the graph convolutional NNGP. When $\nu_\ell \to \infty$ in the DKM objective (Eq. 7) we recover the NNGP, and similarly for the graph DKM objective taking $\nu_\ell \to \infty$ gives us the graph convolutional NNGP. Decreasing the coefficients $\nu_\ell$ decreases the importance of the KL divergence terms, giving Gram matrices flexibility to deviate from the graph convolutional NNGP. In our experiments, we treat the $\nu_\ell$ as knobs for controlling the amount of representation learning. By measuring performance for different $\nu_\ell$, we are able to cleanly demonstrate the importance of learned representations in various tasks.

## 4.2 Inducing-point schemes

In practice it is computationally infeasible to optimize the Gram matrices $\mathbf{G}^1, \ldots, \mathbf{G}^L$ in the graph DKM objective for anything but modestly sized datasets, due to quadratic memory costs of storing the Gram matrices $\mathbf{G}^\ell$, and the cubic time costs. The KL divergence terms, for example, are expensive because of matrix inversions and log determinants:

$$\mathrm{D}_{\mathrm{KL}}\big(\mathcal{N}(\mathbf{0},\mathbf{G}^\ell) \,\|\, \mathcal{N}(\mathbf{0}, \hat{\mathbf{A}}\mathbf{K}(\mathbf{G}^{\ell-1})\hat{\mathbf{A}}^T)\big) =$$
$$\tfrac{1}{2}\big(\mathrm{Tr}((\hat{\mathbf{A}}\mathbf{K}(\mathbf{G}^{\ell-1})\hat{\mathbf{A}}^T)^{-1}\mathbf{G}^\ell) - \log\det((\hat{\mathbf{A}}\mathbf{K}(\mathbf{G}^{\ell-1})\hat{\mathbf{A}}^T)^{-1}\mathbf{G}^\ell) - P\big). \tag{12}$$

We resolve computational issues by developing an inducing point scheme and enabling linear scaling with dataset size. Specifically, we suppose that the graph nodes $\mathbf{X} \in \mathbb{R}^{P \times \nu_0}$ can be summarized sufficiently well by a smaller inducing set $\mathbf{X}_\mathrm{i} \in \mathbb{R}^{P_\mathrm{i} \times \nu_0}$, where the size of the inducing set $P_\mathrm{i}$ is fixed and chosen to be much smaller than the full set of nodes (so that $P_\mathrm{i} \ll P$). These inducing inputs form an initial Gram matrix that we call $\mathbf{G}_{\mathrm{ii}}^0 = \frac{1}{\nu_0}\mathbf{X}_\mathrm{i}\mathbf{X}_\mathrm{i}^T \in \mathbb{R}^{P_\mathrm{i} \times P_\mathrm{i}}$. We also have inducing Gram representations at each layer, $\mathbf{G}_{\mathrm{ii}}^1, \ldots, \mathbf{G}_{\mathrm{ii}}^L$, with all of these inducing representations being learned. The inducing representations are propagated alongside training/test Gram representations $\mathbf{G}_{\mathrm{tt}}^1, \ldots, \mathbf{G}_{\mathrm{tt}}^L$, so that,

$$\mathbf{G}^\ell = \begin{pmatrix} \mathbf{G}_{\mathrm{ii}}^\ell & \mathbf{G}_{\mathrm{it}}^\ell \\ \mathbf{G}_{\mathrm{ti}}^\ell & \mathbf{G}_{\mathrm{tt}}^\ell \end{pmatrix}. \tag{13}$$

The inducing and training/test points interact via the similarity matrix $\mathbf{G}_{\mathrm{ti}}^\ell = (\mathbf{G}_{\mathrm{it}}^\ell)^T \in \mathbb{R}^{P_\mathrm{t} \times P_\mathrm{i}}$, and $\mathbf{G}_{\mathrm{ti}}^\ell$ and $\mathbf{G}_{\mathrm{tt}}^\ell$ are predicted according to our inducing point scheme, in a similar way to how training/test features are predicted in an inducing point GP. We refer readers to Algorithm 1 for computational details.

The form of the inducing Gram matrices depends on the choice of inducing point scheme. The modeller must make a choice as to how the inducing graph nodes interact with the training/test nodes. Broadly, we can either consider intra-domain inducing points, which belong to the same domain as the data, or inter-domain inducing points which do not. We consider one scheme of each type: an intra-domain scheme that assumes adjacency information between inducing points/nodes and training/test nodes, and an inducing scheme that treats the inducing points as unconnected nodes in a graph.

We 'connect' the inducing nodes to each other with an inducing adjacency matrix $\mathbf{A}_{\mathrm{ii}} \in \mathbb{R}^{P_\mathrm{i} \times P_\mathrm{i}}$, and 'connect' the inducing nodes to the training/test nodes with the adjacency $\mathbf{A}_{\mathrm{ti}} \in \mathbb{R}^{P_\mathrm{t} \times P_\mathrm{i}}$. We assume that the adjacency between train/test points is provided in the dataset, which we denote $\mathbf{A}_{\mathrm{tt}}$ for consistency. The two schemes are characterized by,

(1) (intra-domain) sampling a random subset $\mathcal{S} = \{s_1, \ldots, s_{P_\mathrm{i}}\}$ of nodes from the dataset and treating these as inducing points. The inducing adjacencies become $(\mathbf{A}_{\mathrm{ii}})_{jk} = A_{s_j, s_k}$, $(\mathbf{A}_{\mathrm{ti}})_{jk} = A_{j, s_k}$;

(2) (inter-domain) treating inducing nodes as independent of all other nodes, such that $\mathbf{A}_{\mathrm{ii}} = \mathbf{I}$, $\mathbf{A}_{\mathrm{ti}} = \mathbf{0}$.

We give forward propagation rules for each inducing-point scheme, and an ELBO objective (which is used for optimizing the inducing points and replaces the full-rank objective function from Eq. 11) in Appendix B.2. The computational cost of the intra-domain scheme is slightly higher, due to having to store $\mathbf{A}_{\mathrm{ii}}$ and $\mathbf{A}_{\mathrm{ti}}$ and perform matrix multiplications with them. However this cost is negligible compared to storing and computing with the Gram matrices themselves, and further mitigated by the fact that the adjacencies are sparse.

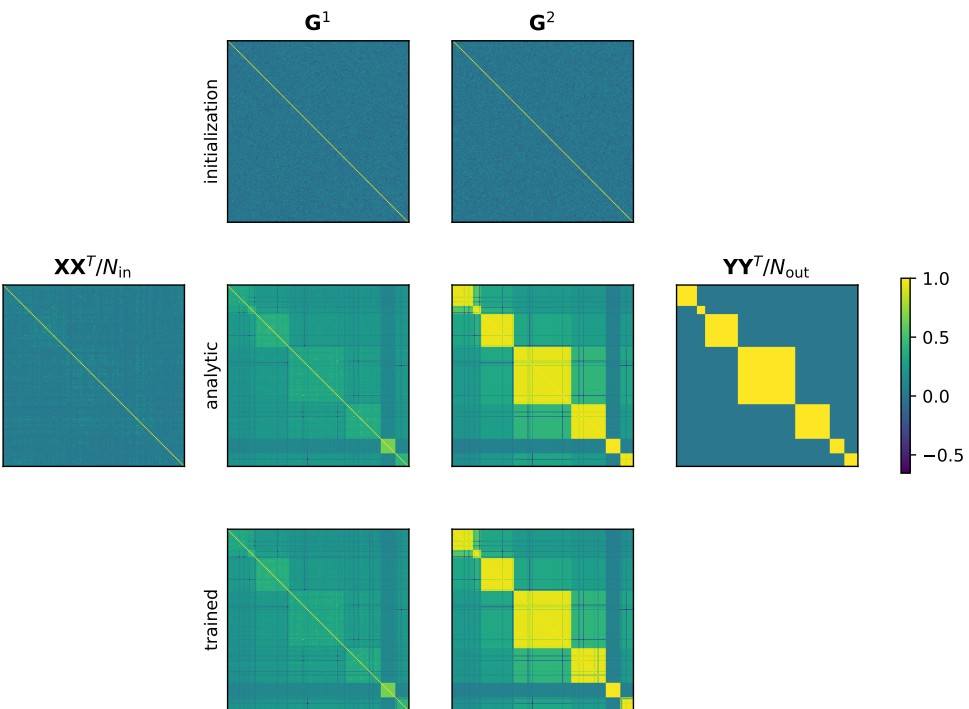

Figure 1: Normalized Gram matrices for a 2-layer linear graph DKM, providing verification of Eq. (14). We use input data, labels, and adjacency from a randomly selected 100 datapoint subset of Cora, and set the adjacency to be $\hat{\mathbf{A}}_{0.5}$. The leftmost and rightmost kernels are the input and label kernels respectively. The middle two columns are the hidden Gram representations. The top row shows Gram matrices after a Wishart initialization, the middle row shows the analytic solution, and the bottom row shows the results after training from this initialization by gradient descent with Adam.

We empirically compare both schemes in Appendix G across several node classification dataset, finding that neither is definitively the best. However, for some datasets (e.g. Citeseer) the inter-domain scheme is markedly better. Given that intra-domain inducing points are not applicable in a multi-graph setting, and that the inter-domain scheme has a simpler implementation, the inter-domain is preferable overall.

## 5 Analysis of Representation Learning in Linear Graph Convolutional Deep Kernel Machines

Remarkably, it is possible to study graph convolutional DKMs analytically in the linear kernel case, which gives insight into representation learning in graphs. Specifically, if we use a linear kernel ($\mathbf{K}(\mathbf{G}) = \mathbf{G}$), set the regularization to 1 ($\nu_\ell = 1$ for all $\ell$), we show in Appendix D that the Gram matrix representations optimizing the graph convolutional DKM objective at each layer $\ell$ are given by,

$$\mathbf{G}^\ell = \hat{\mathbf{A}}^{\ell-1}((\hat{\mathbf{A}}^{-L}\mathbf{G}^{L+1}\hat{\mathbf{A}}^{-L})(\hat{\mathbf{A}}\mathbf{G}^0\hat{\mathbf{A}})^{-1})^{\ell/(L+1)}(\hat{\mathbf{A}}\mathbf{G}^0\hat{\mathbf{A}})\hat{\mathbf{A}}^{\ell-1}. \tag{14}$$

Here $\mathbf{G}^0$ and $\mathbf{G}^{L+1}$ are the input and output kernels respectively (and are assumed to be known). We provide numerical verification of Eq. (14) by comparing it to a solution obtained via gradient descent in Figure 1.

To ensure that the normalized adjacency is invertible, we compute Eq. (14) using $\hat{\mathbf{A}}_\lambda$ in place of $\hat{\mathbf{A}}$, which interpolates with the identity matrix via a parameter $\lambda \in [0, 1]$,

$$\hat{\mathbf{A}}_\lambda = \lambda\mathbf{I} + (1 - \lambda)\hat{\mathbf{A}}. \tag{15}$$

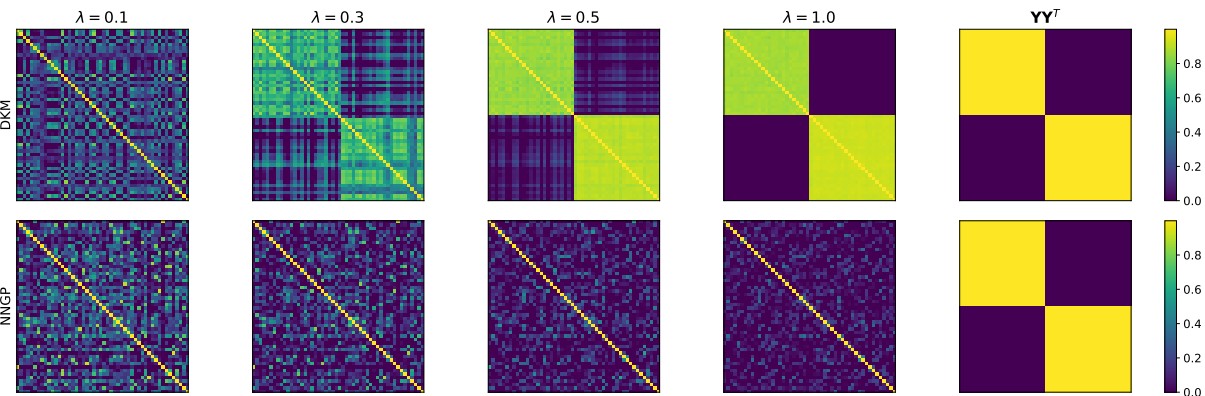

Figure 2: Final layer kernels for 2-layer linear graph convolutional DKMs (top row) and graph convolutional NNGPs (bottom row). The models are fit on a toy dataset with Gaussian random inputs and an Erdős-Rényi adjacency (50 nodes, and edge probability 0.1). The label kernel is shown in the right column. The remaining columns show the effect of varying $\lambda$.

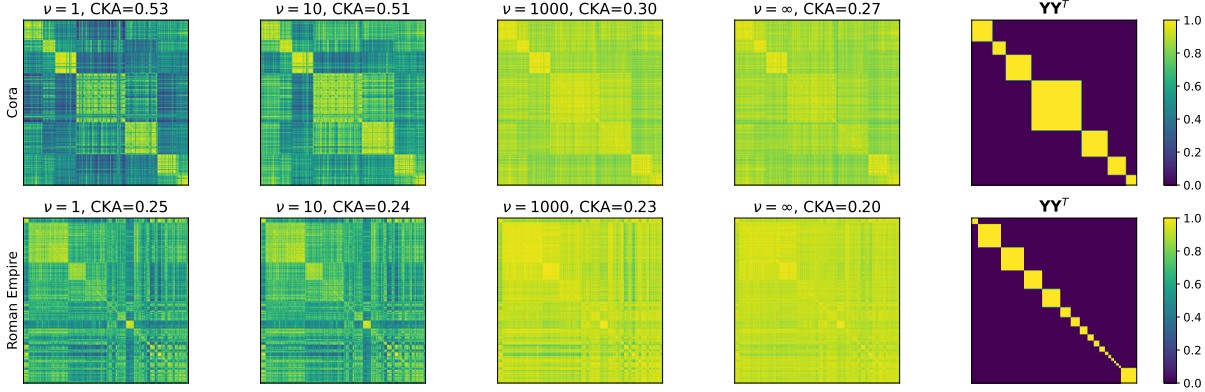

Figure 3: Final layer kernels Cora and Roman Empire kernels for different regularizations $\nu$. The kernels are formed from 400 nodes sampled at random, and the 'true' label kernel is shown on the rightmost column.

It is very natural to introduce this modified adjacency matrix in the graph setting, which incorporated by other prior work (Luan et al., 2020). In particular, with $\lambda = 0$, the graph mixup/convolution performs an equally weighted average over itself and all adjacent nodes. This can be problematic. For instance, consider a network consisting of two connected nodes. After a single graph mixup, the two nodes will have the same features, which is unlikely to be desirable. To fix this issue, we can use $\lambda > 0$, which implies that the graph mixup is now a weighted average, with more weight on self-features. This fixes the issue with our simple example network with two nodes: the resulting features are different even after a graph mixup.

We sought to compare the linear graph convolutional DKM kernel to the linear graph convolutional NNGP kernel,

$$\mathbf{G}_{\text{GCNNGP}}^{\ell} = \hat{\mathbf{A}}^{\ell} \mathbf{G}^0 \hat{\mathbf{A}}^{\ell}. \tag{16}$$

While it is not practical to calculate Eq. (14) for large datasets (due to the inversion of a large adjacency matrix), a comparison is still possible with small datasets. We compare the two kernels on a randomly generated an Erdős-Rényi graph (Erdos & Rényi, 1963) dataset of 50 nodes (with Gaussian features), and insert edges between nodes with probability 0.1. Half of the nodes are given a positive label and the

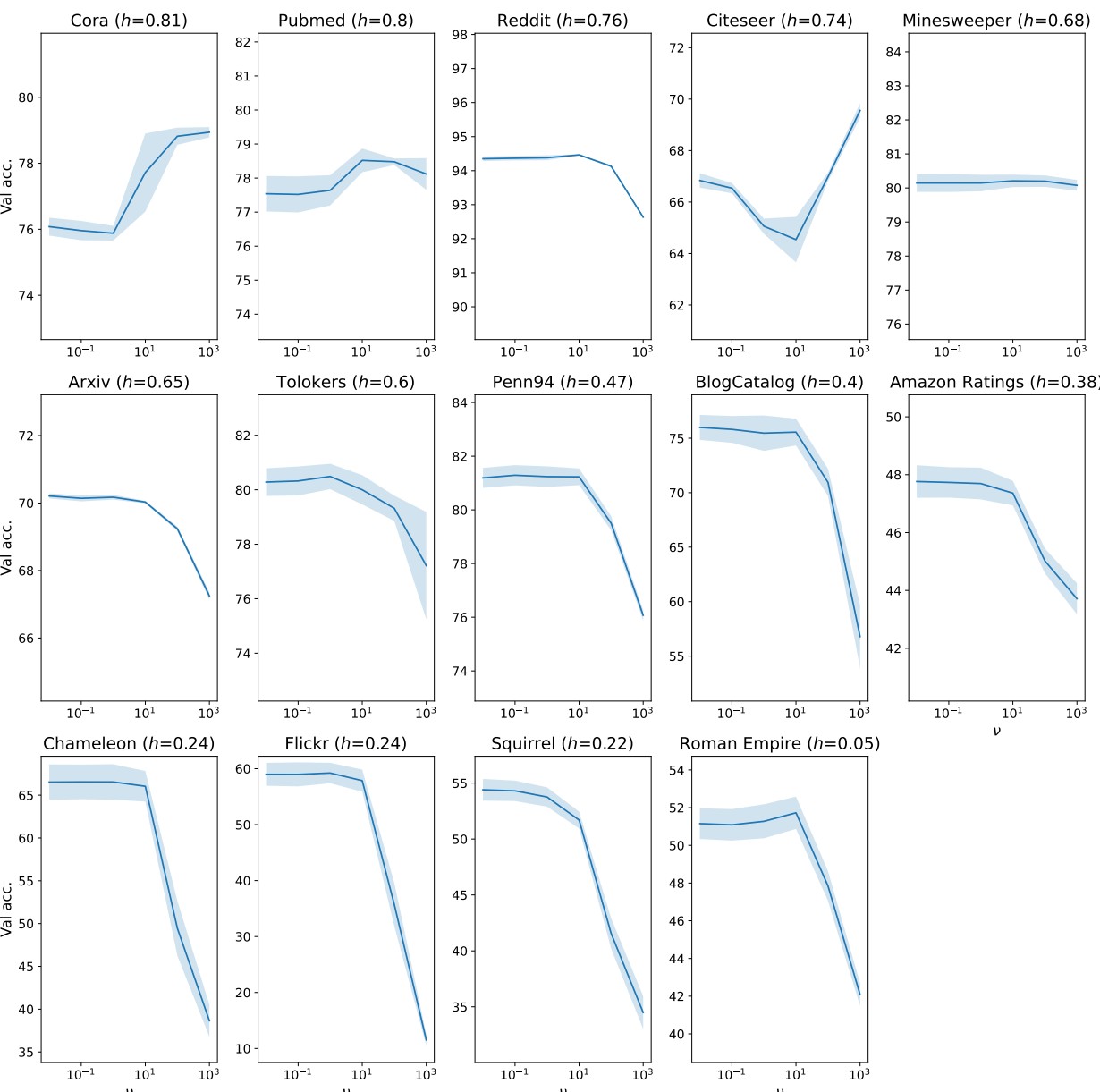

Figure 4: Validation accuracy at different regularization strengths ($\nu$) for each node classification dataset. The datasets are arranged by homophily ratio (denoted by $h$). The error bands denote $\pm 1$ standard deviation in the validation accuracy over several random seeds. For each subplot, we ensure the y-axis range is at least 8, so that the effect of $\nu$ is comparable between dataset.

remainder a negative label. We calculated the output layer kernels for $\lambda \in \{0.1, 0.3, 0.5, 1\}$, and depict results in Figure 2. Note that the Erdős-Rényi adjacency is not homophilous, because an edge is equally likely between both nodes with the same class and with a different class.

We see in Figure 2 (bottom row) that none of the graph convolutional NNGP kernels are aligned to the output kernel, $\mathbf{Y}\mathbf{Y}^T$. This is because the graph convolutional NNGP kernel is a deterministic function of the inputs and adjacency, and is not learned based on the labels, $\mathbf{Y}$. In contrast, the DKM is well-aligned with the output (Figure 2, top row). Note we find that increasing $\lambda$ improves alignment of the graph convolutional

DKM kernels to the output kernel. As $\lambda \to 0$, the adjacency matrix becomes singular, and it seems that this also involves a loss of representation learning, as the graph structure dominates the representation.

## 6 Experiments

In this section we investigate the effect of representation learning in graph networks by training graph convolutional DKMs on several datasets. We train on node classification datasets with varying homophily, as well as graph classification datasets. Our experiments show that representation learning helps to shape output kernels, and that datasets with lower homophily benefit more from representation learning than datasets with high homophily. We also compare the performance of graph convolutional DKMs to GCNs and graph convolutional NNGPs, and find that the extra flexibility afforded by the DKM framework closes the performance gap between graph convolutional NNGPs and GCNs.

In all experiments we use the ReLU/arccosine kernel non-linearity, and set the regularization $\nu_\ell = \nu$ such that it constant for all layers.

### 6.1 Deep Kernel Machines Allow Shaping of Kernel Representations

Similarly to the fully-connected DKM (Yang et al. (2023), Section 4.6), we demonstrate in Figure 3 that the graph convolutional DKM shapes final layer kernel representations towards the label kernel, $\mathbf{Y}\mathbf{Y}^T$. We show alignment both qualitatively via the top-layer kernels, and quantitatively with the Centered Kernel Alignment (CKA) (Cortes et al., 2012) metric. CKA is measured by,

$$\text{CKA}(\mathbf{K}_1, \mathbf{K}_2) = \frac{\text{Tr}(\mathbf{K}_1' \mathbf{K}_2')}{\sqrt{\text{Tr}(\mathbf{K}_1' \mathbf{K}_1') \text{Tr}(\mathbf{K}_2' \mathbf{K}_2')}}, \tag{17}$$

where $\mathbf{K}_{1/2}'$ are centered version of the kernels $\mathbf{K}_{1/2}$ to be compared. Higher CKA values indicate greater similarity between representations, with $\text{CKA}(\mathbf{K}_1, \mathbf{K}_2) \in [0, 1]$.

For this experiment, we trained graph convolutional DKMs for different values of $\nu \in \{10^0, 10^1, 10^3, \infty\}$ on Cora and Roman Empire with the same random seed for 300 epochs. The models were trained using a simple 2-layer architecture with no residual connections or normalization/centering layers, and we used normalized adjacency $\hat{\mathbf{A}}_{\lambda=0.3}$ for the graph convolutions/mixups. The most suitable inducing point scheme was determined by a hyperparameter sweep over schemes for each dataset (see Section 6.2 and Appendix E for details). The kernels shown were obtained by taking the kernel representations of a randomly sampled 400 node subset of the last hidden layer, and the CKA similarity measure was calculated on the full final hidden layer kernel.

When $\nu$ is 1 or 10, we clearly see block-diagonal structure mirroring $\mathbf{Y}\mathbf{Y}^T$, indicating "kernel alignment". As $\nu$ increases to 1000 or $\infty$, flexibility decreases and the kernel alignment appears to become weaker, and is confirmed by the CKA statistics. Interestingly, this breakdown seems to involve all outputs becoming correlated, which is indicative of the rank-collapse-like phenomenon described by Oono & Suzuki (2019).

### 6.2 Investigating the Relationship Between Homophily and Representation Learning

The graph convolutional NNGP kernel has a natural inductive bias for homophilous datasets; the graph convolutional NNGP pools representations from adjacent nodes, therefore in the extreme case that the connected components of the graph correspond to exactly the class labels, we expect its kernel to align with the labels. We expect the graph convolutional NNGP to be less effective for heterophilous datasets, because the graph structure is not informative for the labels. However, representation learning should help in the heterophilous case because we showed in Section 6.1 that increasing flexibility helps shape the kernel representations.

We test the effect of representation learning empirically, training graph convolutional DKMs on several node classification benchmark datasets that exhibit varied levels of homophily. The dataset statistics can be found in Table 3. We trained with different regularization strengths, $\nu \in \{0, 10^{-2}, 10^{-1}, 10^0, 10^1, 10^2, 10^3\}$

Table 1: Mean test accuracies (%) $\pm$ one standard deviation on node classification datasets. Datasets are sorted by homophily, with most homophilous datasets at the top. We provide results for graph convolutional DKMs (GCDKM), graph convolutional NNGPs (GCNNGP) and GCNs. The best accuracy for each dataset is bolded. For datasets with multiple splits, the standard deviation is calculated using the means over splits. In some instances, the best graph convolutional DKM had $\nu = \infty$, which is denoted by ($*$) (hence the sparse GCNNGP results are identical), and ($\dagger$) denotes graph convolutional NNGP results from Niu et al. (2023).

|  | GCDKM | sparse GCNNGP | GCNNGP | GCN (no dropout) | GCN |
|---|---|---|---|---|---|
| Cora | $81.1 \pm 0.2$ | $80.8 \pm 0.1$ | $\mathbf{82.8}^{\dagger}$ | $80.6 \pm 0.2$ | $81.1 \pm 0.3$ |
| Pubmed | $\mathbf{79.8 \pm 0.2}^{*}$ | $\mathbf{79.8 \pm 0.2}$ | $79.6^{\dagger}$ | $79.4 \pm 0.2$ | $79.5 \pm 0.1$ |
| Reddit | $\mathbf{96.2 \pm 0.0}$ | $93.5 \pm 0.0$ | $94.7 \pm 0.0^{\dagger}$ | $95.3 \pm 0.1$ | $95.8 \pm 0.1$ |
| Citeseer | $\mathbf{71.7 \pm 0.4}^{*}$ | $\mathbf{71.7 \pm 0.4}$ | $69.5^{\dagger}$ | $\mathbf{71.7 \pm 0.2}$ | $\mathbf{71.7 \pm 0.2}$ |
| Minesweeper | $\mathbf{85.9 \pm 0.4}$ | $84.3 \pm 0.3$ | — | $85.6 \pm 0.4$ | $85.6 \pm 0.4$ |
| Arxiv | $\mathbf{70.8 \pm 0.2}$ | $68.9 \pm 0.2$ | $70.1 \pm 0.1^{\dagger}$ | $70.0 \pm 0.1$ | $70.6 \pm 0.2$ |
| Tolokers | $81.4 \pm 0.4$ | $79.0 \pm 0.1$ | — | $81.6 \pm 0.7$ | $\mathbf{82.2 \pm 0.7}$ |
| Penn94 | $\mathbf{83.0 \pm 0.2}$ | $75.2 \pm 0.3$ | — | $81.2 \pm 0.5$ | $81.5 \pm 1.8$ |
| BlogCatalog | $92.9 \pm 0.5$ | $92.8 \pm 0.5$ | — | $93.6 \pm 0.6$ | $\mathbf{94.1 \pm 0.6}$ |
| Amazon Ratings | $49.2 \pm 0.5$ | $46.4 \pm 0.3$ | — | $48.5 \pm 0.5$ | $\mathbf{49.8 \pm 0.6}$ |
| Flickr | $\mathbf{85.7 \pm 0.6}$ | $81.5 \pm 2.2$ | — | $81.3 \pm 2.0$ | $82.6 \pm 0.7$ |
| Chameleon | $\mathbf{68.9 \pm 1.4}$ | $65.1 \pm 1.1$ | — | $65.3 \pm 2.6$ | $65.3 \pm 2.6$ |
| Squirrel | $56.8 \pm 1.4$ | $38.6 \pm 1.2$ | — | $57.3 \pm 1.4$ | $\mathbf{57.7 \pm 1.4}$ |
| Roman Empire | $79.2 \pm 0.4$ | $73.6 \pm 0.3$ | — | $79.9 \pm 0.5$ | $\mathbf{82.8 \pm 0.5}$ |

Table 2: Average test accuracies (%) $\pm$ one standard deviation on graph classification datasets. Error bars are not statistically significant. GCN accuracies are sourced from Zhang et al. (2019)($\star$) and Yang et al. (2020) ($\ddagger$).

|  | GCDKM | sparse GCNNGP | GCN |
|---|---|---|---|
| Mutag | $86.6 \pm 3.8$ | $80.8 \pm 5.5$ | $85.6 \pm 5.8^{\ddagger}$ |
| NCI1 | $75.3 \pm 1.0$ | $66.5 \pm 1.0$ | $76.3 \pm 1.8^{\star}$ |
| NCI109 | $75.2 \pm 1.0$ | $65.9 \pm 1.3$ | $75.9 \pm 1.8^{\star}$ |
| Proteins | $72.2 \pm 1.2$ | $67.6 \pm 1.4$ | $75.2 \pm 3.6^{\star}$ |
| Mutagenicity | $79.1 \pm 0.7$ | $72.7 \pm 1.1$ | $79.8 \pm 1.6^{\star}$ |

to control the amount of representation learning. We also trained with two inducing point schemes (detailed in Section 4.2), and for each dataset we selected the optimal scheme by calculating mean validation accuracy for each $\nu$ and scheme. We used a 2-layer architecture, with the adjacency renormalization described by Kipf & Welling (2017), and no residual connections or normalization layers.

Figure 4 illustrates the empirical differences between NNGPs and DKMs, which we find to be highly dataset-dependent. For homophilous datasets (see top row of Figure 4) like Cora, performance tends to remain relatively stable or even improve with larger $\nu$, suggesting that the standard architectural inductive biases are well-suited to these tasks. In contrast, heterophilous datasets such as Roman Empire benefit greatly from smaller $\nu$ (i.e. increased flexibility). This aligns with our theoretical understanding, as DKMs converge to NNGPs in the limit $\nu \to \infty$. The lack of representation learning in NNGPs becomes particularly problematic for small $h$ (bottom row), leading to dramatic declines in accuracy of at least $\sim 10\%$ when comparing $\nu = 10^3$ to $\nu = 10^{-2}$. This suggests that for heterophilous tasks, where standard architectural inductive biases are less appropriate, the graph convolutional DKM's ability to learn representations allows it to compensate, while the graph convolution NNGP's inability to do so results in poor performance.

### 6.3 Final Results

To assess overall performance of the graph convolutional DKM, we optimized hyperparameters for each dataset with a series of grid searches on a range of node and graph classification datasets. We kept the number of layers fixed at 2, and searched over inducing scheme, $\nu$, architecture, kernel centering, and finally the number of inducing points. The final test accuracies are shown in Table 1 and Table 2. We provide details of the datasets and the training procedure in the Appendix E.2.

In Table 1, we see that the graph convolutional DKM performs similarly to the graph convolutional NNGP for homophilous node classification datasets, but less so for heterophilous datasets. We also see that the graph convolutional DKM performs similarly to the GCN for most datasets, closing the performance gap between infinite-width and finite-width networks. We attribute the instances where the GCN outperforms the graph convolutional DKM to dropout, as test accuracy of the graph convolutional DKM is similar to the GCN without dropout. We also provide full-rank graph convolutional NNGP accuracies (where applicable) from Niu et al. (2023). Their graph convolutional NNGP results are slightly different ours, most likely because ours is a sparse model with learned inducing points, whereas theirs is treated like a conventional GP.

In Table 2, we find that graph convolutional DKM performance is competitive with the GCN on graph classifications benchmarks, and uniformly better than the NNGP. The superior performance versus the NNGP is perhaps expected; the graph classification datasets are all molecule datasets, which tend to exhibit a mixture of homophily and heterophily (Ye et al., 2022), and thus the flexibility of the DKM ought to be very helpful.

Overall, we find that the graph convolutional DKM closes the gap between the NNGP and the GCN, supporting the hypothesis that representation learning is the key element lacking from the NNGP. This appears to be true regardless of task (e.g., node classification or graph classification). Further, our results demonstrate that the NNGP is well-suited to homophilous tasks, thus explaining why it is possible to obtain strong NNGP performance on some benchmark graph tasks (Niu et al., 2023), despite NNGPs generally being considered inferior GCNs.

## 7 Limitations

The key limitations of graph convolutional DKMs, like kernel methods in general, is scaling to large dataset sizes due to the cubic cost of naive methods. While we implemented an inducing point scheme to circumvent this limitation, it may be possible to improve on these methods.

Our current analysis considers standard graph convolutional architectures. While it is theoretically possible to develop DKM extensions by 'kernelizing' alternative graph architectures (Bo et al., 2021; Chien et al., 2020; Veličković et al., 2017), these extensions are non-trivial and beyond the scope of this work. We speculate that such extensions would improve performance across all methods (graph convolutional DKM/NNGP, and GCN) by incorporating additional inductive biases. The impact might be most pronounced for NNGPs, which lack representation learning capabilities and thus rely more heavily on built-in inductive biases. In contrast, graph convolutional DKMs and GCNs can compensate for imperfect inductive biases through representation learning.

## 8 Conclusion and Discussion

By leveraging the DKM framework, we have developed a flexible variant of an infinite-width graph convolutional network. The graph convolutional DKM provides a hyperparameter that allows us to interpolate between a flexible kernel machine and the graph convolutional NNGP with fixed kernel. This feature of the DKM enabled us to examine the importance of representation learning on graph node classification tasks, and show competitive performance on graph classification tasks. Remarkably, we found that some tasks benefit more from representation learning than others. This finding helps explain the fact that NNGPs have previously been shown to perform well on graph tasks, but not in other domains. Moreover, the fact that our graph DKM model performs similarly to a GCN without dropout suggests that the kind of flexibility

introduced by the DKM is the sensible kind, and may be used to advance kernel methods in the era of deep learning.

Despite using an inducing point scheme, computational cost of the graph convolutional DKMs is still relatively high. We expect that improving computational efficiency can help improve performance, as it would allow for easier architecture exploration, and training of bigger models. We also expect that more careful optimization could boost performance even further since efficiency gains could enable larger models and longer training. However, we leave these issues for future work.

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

# A  Full Rank Graph Convolutional Deep Kernel Machines

Here, we outline a derivation for the graph convolutional DKM objective. We start with a DGP with graph mixups,

$$\mathrm{P}(\mathbf{F}^\ell \mid \mathbf{F}^{\ell-1}) = \prod_{\lambda=1}^{N_\ell} \mathcal{N}(\mathbf{f}_\lambda^\ell; \mathbf{0}, \hat{\mathbf{A}}\mathbf{K}_{\text{features}}(\mathbf{F}^{\ell-1})\hat{\mathbf{A}}^T), \tag{18}$$

where $\mathbf{f}_\lambda^\ell$ are columns of features,

$$\mathbf{F}^\ell = (\mathbf{f}_1^\ell \cdots \mathbf{f}_{N_\ell}^\ell). \tag{19}$$

We take $P$ to be the number of datapoints, and $N_\ell$ the number of features at each layer $\ell$, so that $\mathbf{F}^\ell \in \mathbb{R}^{P \times N_\ell}$. The log-posterior over Gram matrices $\mathbf{G}^\ell = \frac{1}{N_\ell}\mathbf{F}^\ell(\mathbf{F}^\ell)^T$ is equivalent to,

$$\log \mathrm{P}(\mathbf{G}^1, \ldots, \mathbf{G}^L \mid \mathbf{X}, \mathbf{Y}) = \log \mathrm{P}(\mathbf{Y} \mid \mathbf{G}^L) + \sum_{\ell=1}^{L} \log \mathrm{P}(\mathbf{G}^\ell \mid \mathbf{G}^{\ell-1}), \tag{20}$$

with $\mathbf{G}^0$ the being a kernel formed from the inputs $\mathbf{X}$. Following Section 3.2, we assume that $\mathbf{K}_{\text{features}}(\cdot)$ is a kernel function that can be calculated in terms of Gram matrices, i.e., $\exists \mathbf{K}$ such that $\mathbf{K}(\mathbf{G}^\ell) = \mathbf{K}_{\text{features}}(\mathbf{F}^\ell)$. Examples of suitable kernels are the arccosine kernel (Cho & Saul, 2009) or the squared exponential kernel.

Since $\mathbf{F}^\ell$ is multivariate Gaussian, $\mathbf{G}^\ell$ is Wishart distributed by its definition, and we can write down its log density,

$$\log \mathrm{P}(\mathbf{G}^\ell \mid \mathbf{G}^{\ell-1}) = \frac{N_\ell - P - 1}{2} \log|\mathbf{G}^\ell|$$
$$- \frac{N_\ell}{2}\log|\hat{\mathbf{A}}\mathbf{K}(\mathbf{G}^{\ell-1})\hat{\mathbf{A}}^T| - \frac{N_\ell}{2}\mathrm{Tr}\{(\hat{\mathbf{A}}\mathbf{K}(\mathbf{G}^{\ell-1})\hat{\mathbf{A}}^T)^{-1}\mathbf{G}^\ell\} + \alpha_\ell. \tag{21}$$

The constant $\alpha_\ell$ is the normalizing constant, and we assume that $\hat{\mathbf{A}}$ is full rank. Yang et al. (2023) showed that $\alpha_\ell$ satisfies the following scaling law as we increase width: $\lim_{N_\ell \to \infty} \alpha_\ell / N_\ell = \text{const}$. Therefore if $N_\ell = N\nu_\ell$, it follows that,

$$\lim_{N \to \infty} \frac{1}{N} \log \mathrm{P}(\mathbf{G}^\ell \mid \mathbf{G}^{\ell-1}) = \text{const} + \nu_\ell \mathrm{D}_{\mathrm{KL}}\left(\mathcal{N}(\mathbf{0}, \mathbf{G}^\ell) \,\|\, \mathcal{N}(\mathbf{0}, \hat{\mathbf{A}}\mathbf{K}(\mathbf{G}^{\ell-1})\hat{\mathbf{A}}^T)\right). \tag{22}$$

The Bayesian representation learning limit (Yang et al., 2023) dictates that we scale the final layer alongside the hidden layers (this is unlike the usual NNGP limit (Lee et al., 2017) which only scales the intermediate layers). To do so, we duplicate the outputs so that there are $N$ copies of the labels, i.e. $\tilde{\mathbf{Y}} = (\mathbf{Y} \cdots \mathbf{Y})$. Since channels are assumed to be i.i.d., we have,

$$\mathrm{P}(\tilde{\mathbf{Y}} \mid \mathbf{G}^L) = \mathrm{P}(\mathbf{Y} \mid \mathbf{G}^L)^N. \tag{23}$$

We now consider the posterior of this network with wide hidden layers, wide outputs, and labels $\tilde{\mathbf{Y}}$. Combining Eqs. (20), (22), and (23), and letting $N_\ell = N\nu_\ell$, we have that the scaled log-posterior over Gram matrices satisfies,

$$\lim_{N \to \infty} \frac{1}{N} \log \mathrm{P}(\mathbf{G}^1, \ldots, \mathbf{G}^L \mid \mathbf{X}, \tilde{\mathbf{Y}})$$
$$= \log \mathrm{P}(\mathbf{Y} \mid \mathbf{G}^L) + \sum_{\ell=1}^{L} \nu_\ell \mathrm{D}_{\mathrm{KL}}\left(\mathcal{N}(\mathbf{0}, \mathbf{G}^\ell) \,\|\, \mathcal{N}(\mathbf{0}, \hat{\mathbf{A}}\mathbf{K}(\mathbf{G}^{\ell-1})\hat{\mathbf{A}}^T)\right) + \text{const}$$
$$:= \mathcal{L}(\mathbf{G}^1, \ldots, \mathbf{G}^L). \tag{24}$$

Finally, assuming that the features are sufficiently well-behaved, we note that the Gram matrices are deterministic in the limit $N \to \infty$, since they are a sum of i.i.d. terms,

$$\mathbf{G}^\ell = \frac{1}{N_\ell}\sum_{\lambda=1}^{N_\ell} \mathbf{f}^\ell(\mathbf{f}^\ell)^T \implies \lim_{N \to \infty} \mathbf{G}^\ell = \text{const}. \tag{25}$$

The consequence of Eq. (25) is that the limiting posterior of the Gram matrices is a point distribution. But we know that the Gram matrices on which this point distribution is centered on should maximize the posterior, and hence the scaled log-posterior. Therefore, we seek to maximize Eq. (24), the scaled log-posterior. This is exactly the graph convolutional DKM objective we declared in Section 4.1, up to an additive constant.

## B  Inducing Point Approximations

The graph convolutional DKM objective is difficult to optimize directly due to quadratic memory costs and cubic time complexity. To avoid these large costs, we use an inducing point approximation similar to the one described in (Yang et al., 2023). We review in this Appendix how to construct an inducing point objective in a fully-connect setting, and then extend to the graph setting.

### B.1  An Inducing Point Objective

Sparse DKMs as derived by Yang et al. (2023) are inspired by the sparse DGP literature (Damianou & Lawrence, 2013; Salimbeni & Deisenroth, 2017). In essence, we place an approximate posterior on a set of sparse inducing-points/features and derive an objective to optimize these inducing points by finding a lower bound on the evidence with variational inference (VI). In this subsection we describe the procedure.

We introducing some inducing-points/features $\mathbf{F}_i^\ell \in \mathbb{R}^{P_i \times N_\ell}$ in addition to the 'real' test/training features $\mathbf{F}_t^\ell \in \mathbb{R}^{P_t \times N_\ell}$ at each layer in the following prior model,

$$\mathbf{F}^\ell := \begin{pmatrix} \mathbf{F}_i^\ell \\ \mathbf{F}_t^\ell \end{pmatrix}, \tag{26a}$$

$$\mathrm{P}(\mathbf{F}^\ell \mid \mathbf{F}^{\ell-1}) = \prod_{\lambda=1}^{N_\ell} \mathcal{N}(\mathbf{f}_\lambda^\ell; \mathbf{0}, \mathbf{K}_{\mathrm{features}}(\mathbf{F}^{\ell-1})), \tag{26b}$$

$$\mathrm{P}(\mathbf{W}) = \prod_{\lambda=1}^{\nu_{L+1}} \mathcal{N}(\mathbf{w}_\lambda; \mathbf{0}, \mathbf{I}), \tag{26c}$$

$$\mathrm{P}(\mathbf{Y} \mid \mathbf{K}^L, \mathbf{W}) = \prod_{i=1}^{P_t} \mathrm{Categorical}(\mathbf{y}_i; \mathrm{softmax}(\mathbf{K}_{ti}^L \mathrm{chol}(\mathbf{K}_{ii}^L)^{-1}\mathbf{W})_i), \tag{26d}$$

where $\mathbf{K}^L$ is the kernel applied to the final hidden layer features $\mathbf{F}^L$, with $\mathbf{K}_{ii}^L$ being the inducing/inducing sub-block and $\mathbf{K}_{ti}^L$ being the training/test-inducing sub-block. Eq. (26) is similar to the DGP in Eq. (1), but it has a different top-layer. The motivation for using the top-layer in Eq. (26) is efficiency; it is more efficient than a categorical GP under the parameterization described in Appendix I.

To perform variational inference, we place a Gaussian approximate posterior on the inducing hidden-layer features,

$$\mathrm{Q}(\mathbf{F}_i^\ell) = \prod_{\lambda=1}^{N_\ell} \mathcal{N}((\mathbf{f}_i^\ell)_\lambda; \mathbf{0}, \mathbf{G}_{ii}), \tag{27}$$

and the following approximate posterior on the top-layer,

$$\mathrm{Q}(\mathbf{W}) = \prod_{\lambda=1}^{\nu_{L+1}} \mathcal{N}(\mathbf{w}_\lambda; \boldsymbol{\mu}_\lambda, \boldsymbol{\Sigma}). \tag{28}$$

Thus, $\mathbf{G}_{ii}^1, \ldots, \mathbf{G}_{ii}^L, \boldsymbol{\mu}_1, \ldots, \boldsymbol{\mu}_{\nu_{L+1}}$, and $\boldsymbol{\Sigma}$ are the variational parameters. We can write down the distribution of the test/training features conditional on the inducing features using standard multivariate Gaussian rules,

$$\mathrm{P}(\mathbf{F}_t^\ell \mid \mathbf{F}_i^\ell, \mathbf{F}^{\ell-1}) = \mathcal{N}\left(\mathbf{F}_t^\ell; \mathbf{K}_{ti}\mathbf{K}_{ii}^{-1}\mathbf{F}_i^\ell, \mathbf{K}_{tt} - \mathbf{K}_{ti}\mathbf{K}_{ii}^{-1}\mathbf{K}_{it}\right), \tag{29}$$

where by $\mathbf{K}_{ii}$, $\mathbf{K}_{ti}$, and $\mathbf{K}_{tt}$ we refer to inducing-inducing, test/train-inducing, and test/train-test/train blocks respectively of $\mathbf{K}(\mathbf{F}^{\ell-1})$. We use Eq. (29) to construct an approximate posterior over the test/training features. The prior at each layer is equivalent to,

$$\mathrm{P}(\mathbf{F}^{\ell} \mid \mathbf{F}^{\ell-1}) = \mathrm{P}(\mathbf{F}_{t}^{\ell} \mid \mathbf{F}_{i}^{\ell}, \mathbf{F}^{\ell-1}) \, \mathrm{P}(\mathbf{F}_{i}^{\ell} \mid \mathbf{F}_{i}^{\ell-1}), \tag{30}$$

and we select the full approximate posterior at each hidden layer to be

$$\mathrm{Q}(\mathbf{F}^{\ell} \mid \mathbf{F}^{\ell-1}) = \mathrm{P}(\mathbf{F}_{t}^{\ell} \mid \mathbf{F}_{i}^{\ell}, \mathbf{F}^{\ell-1}) \, \mathrm{Q}(\mathbf{F}_{i}^{\ell}). \tag{31}$$

This choice of approximate posterior is a classic one; the common term, $\mathrm{P}(\mathbf{F}_{t}^{\ell} \mid \mathbf{F}_{i}^{\ell}, \mathbf{F}^{\ell-1})$, is going to lead to cancellation in the ELBO. At the top layer, we use an i.i.d. Gaussian approximate posterior for the weights $\mathbf{W}$,

$$\mathrm{Q}(\mathbf{W}) = \prod_{\lambda=1}^{\nu_{L+1}} \mathcal{N}(\mathbf{w}_{\lambda}; \boldsymbol{\mu}_{\lambda}, \boldsymbol{\Sigma}). \tag{32}$$

Similarly to Eq. (23) in the full-rank derivation, we duplicate the entries in the top-layer $N$ times and make an i.i.d. assumption, such that $\tilde{\mathbf{Y}} = (\mathbf{Y} \cdots \mathbf{Y}) \in \mathbb{R}^{P_t \times N\nu_{L+1}}$ and $\tilde{\mathbf{W}} = (\mathbf{W} \cdots \mathbf{W}) \in \mathbb{R}^{P_i \times N\nu_{L+1}}$. Note that both the prior and the approximate posterior factorize across layers, which allows us to simplify the ELBO,

$$\mathrm{ELBO} = \mathbb{E}_{\mathrm{Q}} \left[ \log \frac{\mathrm{P}(\tilde{\mathbf{Y}}, \mathbf{X}, \mathbf{F}^1, \ldots, \mathbf{F}^L, \tilde{\mathbf{W}})}{\mathrm{Q}(\mathbf{F}^1, \ldots, \mathbf{F}^L, \tilde{\mathbf{W}})} \right] \tag{33a}$$

$$= \mathbb{E}_{\mathrm{Q}} \left[ \log \frac{\mathrm{P}(\tilde{\mathbf{Y}}, \tilde{\mathbf{W}} \mid \mathbf{F}^L)}{\mathrm{Q}(\tilde{\mathbf{W}})} + \sum_{\ell=1}^{L} \log \frac{\mathrm{P}(\mathbf{F}^{\ell} \mid \mathbf{F}^{\ell-1})}{\mathrm{Q}(\mathbf{F}^{\ell})} \right] \tag{33b}$$

$$= \mathbb{E}_{\mathrm{Q}} \left[ \log \frac{\mathrm{P}(\tilde{\mathbf{Y}}, \tilde{\mathbf{W}} \mid \mathbf{K}^L)}{\mathrm{Q}(\tilde{\mathbf{W}})} \right] + \sum_{\ell=1}^{L} \mathbb{E}_{\mathrm{Q}} \left[ \log \frac{\mathrm{P}(\mathbf{F}_{i}^{\ell} \mid \mathbf{F}_{i}^{\ell-1})}{\mathrm{Q}(\mathbf{F}_{i}^{\ell})} \right]. \tag{33c}$$

In the limit $N \to \infty$, where $N_{\ell} = N\nu_{\ell}$, the covariances at each layer, $N_{\ell}^{-1}\mathbf{F}_{i}^{\ell}(\mathbf{F}_{i}^{\ell})^T$, collapse to become equal to $\mathbf{G}_{ii}^{\ell}$. It can be shown that the hidden-layer terms in the ELBO simplify,

$$\frac{1}{N}\mathbb{E}_{\mathrm{Q}} \left[ \log \frac{\mathrm{P}(\mathbf{F}_{i}^{\ell} \mid \mathbf{F}_{i}^{\ell-1})}{\mathrm{Q}(\mathbf{F}_{i}^{\ell})} \right] \to -\nu_{\ell}\mathrm{D}_{\mathrm{KL}}\left( \mathcal{N}(\mathbf{0}, \mathbf{G}_{ii}^{\ell}) \,\|\, \mathcal{N}(\mathbf{0}, \mathbf{K}(\mathbf{G}_{ii}^{\ell-1})) \right). \tag{34}$$

Additionally, $\forall N$ the term involving $\tilde{\mathbf{Y}}$ satisfies,

$$\frac{1}{N}\mathbb{E}_{\mathrm{Q}} \left[ \log \frac{\mathrm{P}(\tilde{\mathbf{Y}}, \tilde{\mathbf{W}} \mid \mathbf{K}^L)}{\mathrm{Q}(\tilde{\mathbf{W}})} \right] = \mathbb{E}_{\mathrm{Q}} \left[ \log \mathrm{P}(\mathbf{Y} \mid \mathbf{K}^L, \mathbf{W}) + \log \frac{\mathrm{P}(\mathbf{W})}{\mathrm{Q}(\mathbf{W})} \right] \tag{35a}$$

$$= \mathbb{E}_{\mathrm{Q}}[\log \mathrm{P}(\mathbf{Y} \mid \mathbf{K}^L, \mathbf{W})] - \sum_{\lambda=1}^{\nu_{L+1}} \mathrm{D}_{\mathrm{KL}}\left( \mathcal{N}(\boldsymbol{\mu}_{\lambda}, \boldsymbol{\Sigma}) \,\|\, \mathcal{N}(\mathbf{0}, \mathbf{I}) \right), \tag{35b}$$

which can easily be estimated by sampling $\mathbf{W}$, since in the limit the kernel $\mathbf{K}^L = \mathbf{K}(\mathbf{G}^L)$ is constant.

We define the sparse DKM objective to be the limiting ELBO scaled by $N^{-1}$,

$$\lim_{N \to \infty} \frac{1}{N}\mathrm{ELBO} := \mathcal{L}_{\mathrm{sparse}} = \mathbb{E}_{\mathrm{Q}}[\log \mathrm{P}(\mathbf{Y} \mid \mathbf{K}(\mathbf{G}^L), \mathbf{W})]$$
$$- \sum_{\lambda=1}^{\nu_{L+1}} \mathrm{D}_{\mathrm{KL}}\left( \mathcal{N}(\boldsymbol{\mu}_{\lambda}, \boldsymbol{\Sigma}) \,\|\, \mathcal{N}(\mathbf{0}, \mathbf{I}) \right)$$
$$- \sum_{\ell=1}^{L} \nu_{\ell}\mathrm{D}_{\mathrm{KL}}\left( \mathcal{N}(\mathbf{0}, \mathbf{G}_{ii}^{\ell}) \,\|\, \mathcal{N}(\mathbf{0}, \mathbf{K}(\mathbf{G}_{ii}^{\ell-1})) \right). \tag{36}$$

Notice that the likelihood term in Eq. (36) requires the full kernel. We calculate this recursively. Eq. (29) tells us that,

$$\mathbf{F}_{\mathrm{t}}^{\ell} = \mathbf{K}_{\mathrm{ti}}\mathbf{K}_{\mathrm{ii}}^{-1}\mathbf{F}_{\mathrm{i}}^{\ell} + (\mathbf{K}_{\mathrm{tt}} - \mathbf{K}_{\mathrm{ti}}\mathbf{K}_{\mathrm{ii}}^{-1}\mathbf{K}_{\mathrm{it}})^{1/2}\boldsymbol{\Xi}, \tag{37}$$

where $\boldsymbol{\Xi}$ is a matrix of i.i.d. standard Gaussian noise, and $\mathbf{K} = \mathbf{K}(\mathbf{G}^{\ell-1}) = [\mathbf{K}_{\mathrm{ii}} \ \mathbf{K}_{\mathrm{it}}; \ \mathbf{K}_{\mathrm{ti}} \ \mathbf{K}_{\mathrm{tt}}]$. Therefore,

$$\mathbf{G}_{\mathrm{ti}}^{\ell} = \lim N_{\ell}^{-1}\mathbf{F}_{\mathrm{t}}^{\ell}(\mathbf{F}_{\mathrm{i}}^{\ell})^{T} = \mathbf{K}_{\mathrm{ti}}\mathbf{K}_{\mathrm{ii}}^{-1}\mathbf{G}_{\mathrm{ii}}^{\ell}, \tag{38}$$

$$\mathbf{G}_{\mathrm{tt}}^{\ell} = \lim N_{\ell}^{-1}\mathbf{F}_{\mathrm{t}}^{\ell}(\mathbf{F}_{\mathrm{t}}^{\ell})^{T} = \mathbf{K}_{\mathrm{tt}} - \mathbf{K}_{\mathrm{ti}}\mathbf{K}_{\mathrm{ii}}^{-1}\mathbf{K}_{\mathrm{it}} + \mathbf{K}_{\mathrm{ti}}\mathbf{K}_{\mathrm{ii}}^{-1}\mathbf{G}_{\mathrm{ii}}^{\ell}\mathbf{K}_{\mathrm{ii}}^{-1}\mathbf{K}_{\mathrm{it}}. \tag{39}$$

### B.2 Sparse Graph Convolutional Deep Kernel Machines

To use a graph convolutional DKM, we need to be able to compute the graph kernel $\mathbf{G} \mapsto \mathbf{A}\mathbf{K}(\mathbf{G})\mathbf{A}^{T}$ with our inducing points. We already have adjacency information regarding the training/test nodes, but not the inducing nodes. Therefore we assume we have an adjacency between inducing points, $\mathbf{A}_{\mathrm{ii}}$, as well as adjacency information between inducing points and training/test points, $\mathbf{A}_{\mathrm{it}} = \mathbf{A}_{\mathrm{ti}}^{T}$. For notational consistency, we denote the adjacency of the data as $\mathbf{A}_{\mathrm{tt}}$.

The graph convolutional kernel over Gram matrices when using an inducing scheme becomes

$$\mathbf{K}_{\mathrm{GC}}(\mathbf{G}) = \begin{pmatrix} \mathbf{A}_{\mathrm{ii}} & \mathbf{A}_{\mathrm{it}} \\ \mathbf{A}_{\mathrm{ti}} & \mathbf{A}_{\mathrm{tt}} \end{pmatrix} \mathbf{K}(\mathbf{G}) \begin{pmatrix} \mathbf{A}_{\mathrm{ii}} & \mathbf{A}_{\mathrm{it}} \\ \mathbf{A}_{\mathrm{ti}} & \mathbf{A}_{\mathrm{tt}} \end{pmatrix}^{T}. \tag{40}$$

where $\mathbf{K}(\mathbf{G})$ is a "base kernel" (e.g. the arccosine kernel) applied to $\mathbf{G}$. We derive this kernel for an graph convolutional NNGP concretely in Appendix C. Substituting this graph kernel into the intermediate layers of the sparse DKM in Eq. (36) gives us a sparse graph convolutional DKM objective,

$$\begin{aligned} \mathcal{L}_{\mathrm{GC\text{-}sparse}} = \ &\mathbb{E}_{\mathbf{W} \sim \mathrm{Q}}[\log \mathrm{P}(\mathbf{Y} \mid \mathbf{K}(\mathbf{G}^{L}), \mathbf{W})] \\ &- \sum_{\lambda=1}^{\nu_{L+1}} \mathrm{D}_{\mathrm{KL}}\left(\mathcal{N}(\boldsymbol{\mu}_{\lambda}, \boldsymbol{\Sigma}) \,\|\, \mathcal{N}(\mathbf{0}, \mathbf{I})\right) \\ &- \sum_{\ell=1}^{L} \nu_{\ell}\mathrm{D}_{\mathrm{KL}}\left(\mathcal{N}(\mathbf{0}, \mathbf{G}_{\mathrm{ii}}^{\ell}) \,\|\, \mathcal{N}(\mathbf{0}, \mathbf{A}_{\mathrm{ii}}\mathbf{K}(\mathbf{G}_{\mathrm{ii}}^{\ell-1})\mathbf{A}_{\mathrm{ii}}^{T})\right), \end{aligned} \tag{41}$$

Again, the Gram matrices are calculated recursively in the forward pass with,

$$\mathbf{G}_{\mathrm{ti}}^{\ell} = \mathbf{K}_{\mathrm{ti}}\mathbf{K}_{\mathrm{ii}}^{-1}\mathbf{G}_{\mathrm{ii}}^{\ell}, \tag{42}$$

$$\mathbf{G}_{\mathrm{tt}}^{\ell} = \mathbf{K}_{\mathrm{tt}} - \mathbf{K}_{\mathrm{ti}}\mathbf{K}_{\mathrm{ii}}^{-1}\mathbf{K}_{\mathrm{it}} + \mathbf{K}_{\mathrm{ti}}\mathbf{K}_{\mathrm{ii}}^{-1}\mathbf{G}_{\mathrm{ii}}^{\ell}\mathbf{K}_{\mathrm{ii}}^{-1}\mathbf{K}_{\mathrm{it}}, \tag{43}$$

and we then apply the graph convolutional kernel $\mathbf{K}_{\mathrm{GC}}(\cdot)$ to $\mathbf{G}^{\ell} = [\mathbf{G}_{\mathrm{ii}}^{\ell} \ \mathbf{G}_{\mathrm{it}}^{\ell}; \ \mathbf{G}_{\mathrm{ti}}^{\ell} \ \mathbf{G}_{\mathrm{tt}}^{\ell}]$.

We considered two choices for $\mathbf{A}_{\mathrm{ii}}$ and $\mathbf{A}_{\mathrm{ti}}/\mathbf{A}_{\mathrm{it}}$, namely

(1) (intra-domain) using adjacency information from inducing points sampled from the training set,

(2) (inter-domain) treating inducing points as independent of other inducing points and the training/test points, i.e. $\mathbf{A}_{\mathrm{ii}} = \mathbf{I}$, $\mathbf{A}_{\mathrm{ti}} = \mathbf{0}$.

In the case where the dataset is a single graph, we use a minibatch size of 1, and the 'minibatch' becomes all nodes in the graph. In the case where the dataset contains several graphs, the graphs can be batched together. Naively, this would involve creating a single large adjacency matrix that describes adjacency information for graphs in the minibatch, but in practice this can be efficiently represented with a sparse tensor. The inter-domain scheme has the advantage that it can be used for datasets with a single graph or multiple graphs, whereas the intra-domain scheme is not expected to work when we change the graph batch (in the multi-graph case).

## C  Graph Convolutional NNGP Kernel Derivations

We show how to derive the kernel/covariance expressions given in Eq. (40) (for the sparse inducing point case). Eq. (8) (for the full-rank case) is the same, but with inducing points dropped.

We take the graph nodes to have adjacency information as described in Section 4.2 and Appendix B. In a graph convolutional NNGP, pre-activation features $\mathbf{F}_i^\ell, \mathbf{F}_t^\ell$ for the inducing and test/train blocks can be written in terms of the post-activations $\mathbf{H}_i^{\ell-1}$, $\mathbf{H}_t^{\ell-1}$,

$$\begin{pmatrix} \mathbf{F}_i^\ell \\ \mathbf{F}_t^\ell \end{pmatrix} = \begin{pmatrix} \mathbf{A}_{ii} & \mathbf{A}_{it} \\ \mathbf{A}_{ti} & \mathbf{A}_{tt} \end{pmatrix} \begin{pmatrix} \mathbf{H}_i^{\ell-1} \\ \mathbf{H}_t^{\ell-1} \end{pmatrix} \mathbf{W}, \tag{44}$$

where $\mathbf{W} \in \mathbb{R}^{N_{\ell-1} \times N_\ell}$ has i.i.d. Gaussian elements with mean zero and variance $1/N_{\ell-1}$.

Remember that each column $\lambda$ of $\mathbf{F}^\ell$ is i.i.d., so the covariance of $\mathbf{F}^\ell$ is,

$$\begin{pmatrix} \mathbf{K}_{ii} & \mathbf{K}_{it} \\ \mathbf{K}_{ti} & \mathbf{K}_{tt} \end{pmatrix} = \mathbb{E}\left[ \begin{pmatrix} \mathbf{f}_\lambda^{i;\ell} \\ \mathbf{f}_\lambda^{t;\ell} \end{pmatrix} \begin{pmatrix} \mathbf{f}_\lambda^{i;\ell} \\ \mathbf{f}_\lambda^{t;\ell} \end{pmatrix}^T \right] \tag{45a}$$

$$= \begin{pmatrix} \mathbf{A}_{ii} & \mathbf{A}_{it} \\ \mathbf{A}_{ti} & \mathbf{A}_{tt} \end{pmatrix} \begin{pmatrix} \mathbf{H}_i^{\ell-1} \\ \mathbf{H}_t^{\ell-1} \end{pmatrix} \mathbb{E}[\mathbf{w}_\lambda \mathbf{w}_\lambda^T] \begin{pmatrix} \mathbf{H}_i^{\ell-1} \\ \mathbf{H}_t^{\ell-1} \end{pmatrix}^T \begin{pmatrix} \mathbf{A}_{ii} & \mathbf{A}_{it} \\ \mathbf{A}_{ti} & \mathbf{A}_{tt} \end{pmatrix}^T \tag{45b}$$

$$= \begin{pmatrix} \mathbf{A}_{ii} & \mathbf{A}_{it} \\ \mathbf{A}_{ti} & \mathbf{A}_{tt} \end{pmatrix} \left[ \frac{1}{N_{\ell-1}} \begin{pmatrix} \mathbf{H}_i^{\ell-1} \\ \mathbf{H}_t^{\ell-1} \end{pmatrix} \begin{pmatrix} \mathbf{H}_i^{\ell-1} \\ \mathbf{H}_t^{\ell-1} \end{pmatrix}^T \right] \begin{pmatrix} \mathbf{A}_{ii} & \mathbf{A}_{it} \\ \mathbf{A}_{ti} & \mathbf{A}_{tt} \end{pmatrix}^T . \tag{45c}$$

This is the general form of the graph convolutional kernel. However, if the activation function is ReLU, i.e. $\mathbf{H}^\ell = \mathrm{ReLU}(\mathbf{F}^\ell)$, then in the limit $N_{\ell-1} \to \infty$, $N_{\ell-1}^{-1} \mathbf{H}^{\ell-1}(\mathbf{H}^{\ell-1})^T \to \mathbf{\Phi}(\mathbf{K}^{\ell-1})$, where $\mathbf{\Phi}(\cdot)$ is arccos kernel (Cho & Saul, 2009) and $\mathbf{K}^{\ell-1}$ is the previous layer's kernel. This gives the recursive relation

$$\begin{pmatrix} \mathbf{K}_{ii}^\ell & \mathbf{K}_{it}^\ell \\ \mathbf{K}_{ti}^\ell & \mathbf{K}_{tt}^\ell \end{pmatrix} = \begin{pmatrix} \mathbf{A}_{ii} & \mathbf{A}_{it} \\ \mathbf{A}_{ti} & \mathbf{A}_{tt} \end{pmatrix} \mathbf{\Phi}\left( \begin{pmatrix} \mathbf{K}_{ii}^{\ell-1} & \mathbf{K}_{it}^{\ell-1} \\ \mathbf{K}_{ti}^{\ell-1} & \mathbf{K}_{tt}^{\ell-1} \end{pmatrix} \right) \begin{pmatrix} \mathbf{A}_{ii} & \mathbf{A}_{it} \\ \mathbf{A}_{ti} & \mathbf{A}_{tt} \end{pmatrix}^T . \tag{46}$$

We call this kernel the graph convolutional kernel.

### C.1  Sparse Graph Convolutional Deep Kernel Machine Algorithm

For completeness, we provide a Algorithm 1 to demonstrate node prediction with the sparse graph convolutional DKM. We discuss how to parameterize the learned inducing points for training in Appendix I. The algorithm can be modified for graph classification by adding a mean-pool layer immediately before sampling.

For the 'sample' step at the top-layer, we have two main options. The first, most conventional method, is to sample with a sparse GP (similar to Milsom et al. (2023)). The second, which is the approach taken in this paper, is to sample weights $\mathbf{W}$ from the approximate posterior (Eq. 28), which in turn allows sampling logits by calculating $\mathbf{K}_{ti}\mathrm{chol}(\mathbf{K}_{ii})^{-1}\mathbf{W} \in \mathbb{R}^{P_t \times \nu_{L+1}}$.

---

**Algorithm 1** Graph convolutional DKM node classification

---

**Parameters:** $\{\nu_\ell\}_{\ell=1}^{L}$
**Train/test inputs, train labels:** $\mathbf{X}_t$, $\mathbf{Y}_t$
**Inducing and train/test adjacencies:** $\mathbf{A}_{ii}$, $\mathbf{A}_{ti}$, $\mathbf{A}_{tt}$
**Inducing inputs and inducing Gram matrices:** $\mathbf{X}_i$, $\{\mathbf{G}_{ii}^{\ell}\}_{\ell=1}^{L+1}$
**Variational parameters for the output layer:** $\boldsymbol{\mu}_1, \ldots, \boldsymbol{\mu}_{\nu_{L+1}}, \boldsymbol{\Sigma}$
Initialize full Gram matrix
$$\begin{pmatrix} \mathbf{G}_{ii}^{0} & \mathbf{G}_{it}^{0} \\ \mathbf{G}_{ti}^{0} & \mathbf{G}_{tt}^{0} \end{pmatrix} \leftarrow \frac{1}{\nu_0} \begin{pmatrix} \mathbf{X}_i\mathbf{X}_i^{T} & \mathbf{X}_i\mathbf{X}_t^{T} \\ \mathbf{X}_t\mathbf{X}_i^{T} & \mathbf{X}_t\mathbf{X}_t^{T} \end{pmatrix}$$
**for** $\ell$ in $(1, \ldots, L)$ **do**
  Apply kernel non-linearity $\boldsymbol{\Phi}$ and perform graph convolution
  $$\begin{pmatrix} \mathbf{K}_{ii} & \mathbf{K}_{it} \\ \mathbf{K}_{ti} & \mathbf{K}_{tt} \end{pmatrix} \leftarrow \begin{pmatrix} \mathbf{A}_{ii} & \mathbf{A}_{it} \\ \mathbf{A}_{ti} & \mathbf{A}_{tt} \end{pmatrix} \boldsymbol{\Phi}\left(\begin{pmatrix} \mathbf{G}_{ii}^{\ell-1} & \mathbf{G}_{it}^{\ell-1} \\ \mathbf{G}_{ti}^{\ell-1} & \mathbf{G}_{tt}^{\ell-1} \end{pmatrix}\right) \begin{pmatrix} \mathbf{A}_{ii} & \mathbf{A}_{it} \\ \mathbf{A}_{ti} & \mathbf{A}_{tt} \end{pmatrix}^{T}$$
  Propagate the test-test and test-inducing blocks
  $$\mathbf{G}_{ti}^{\ell} \leftarrow \mathbf{K}_{ti}\mathbf{K}_{ii}^{-1}\mathbf{G}_{ii}^{\ell}$$
  $$\mathbf{G}_{tt}^{\ell} \leftarrow \mathbf{K}_{tt} - \mathbf{K}_{ti}\mathbf{K}_{ii}^{-1}\mathbf{K}_{it} + \mathbf{K}_{ti}\mathbf{K}_{ii}^{-1}\mathbf{G}_{ii}^{\ell}\mathbf{K}_{ii}^{-1}\mathbf{K}_{it}$$
**end for**
Calculate output kernel
$$\begin{pmatrix} \mathbf{K}_{ii} & \mathbf{K}_{it} \\ \mathbf{K}_{ti} & \mathbf{K}_{tt} \end{pmatrix} \leftarrow \boldsymbol{\Phi}\left(\begin{pmatrix} \mathbf{G}_{ii}^{L} & \mathbf{G}_{it}^{L} \\ \mathbf{G}_{ti}^{L} & \mathbf{G}_{tt}^{L} \end{pmatrix}\right)$$
Perform prediction using a sparse GP (or alternative model) with output kernel
$$\hat{\mathbf{Y}} \leftarrow \text{ sample } \mathrm{P}(\mathbf{Y} \mid \mathbf{K}_{ii}, \mathbf{K}_{ti}, \mathbf{K}_{tt})$$

---

# D    Closed-form Solution for a Linear Graph Convolutional Deep Kernel Machine

For some node regression problems with a linear kernel, $\mathbf{K}(\mathbf{G}) = \mathbf{G}$, we are able to find an optimum for the DKM analytically. Assuming a constant regularizer coefficient for each layer, $\nu_\ell = 1$, and no observation noise, the objective $\mathcal{L}$ for a linear graph convolutional DKM is,

$$\mathcal{L}(\mathbf{G}^1, \ldots, \mathbf{G}^L) = -\sum_{\ell=1}^{L+1} \mathrm{D}_{\mathrm{KL}}(\mathcal{N}(\mathbf{0}, \mathbf{G}^{\ell}) \,||\, \mathcal{N}(\mathbf{0}, \hat{\mathbf{A}}\mathbf{K}(\mathbf{G}^{\ell-1})\hat{\mathbf{A}}^{T})) \tag{47a}$$

$$= \text{const} + \frac{\nu}{2}\sum_{\ell=1}^{L+1}\left[\log|(\hat{\mathbf{A}}\mathbf{G}^{\ell-1}\hat{\mathbf{A}}^{T})^{-1}\mathbf{G}^{\ell}| - \mathrm{Tr}((\hat{\mathbf{A}}\mathbf{G}^{\ell-1}\hat{\mathbf{A}}^{T})^{-1}\mathbf{G}^{\ell})\right]. \tag{47b}$$

A slight abuse of notation follows, in that superscripts $\ell$ on $\mathbf{G}^{\ell}$ and $\mathbf{Z}^{\ell}$ refer to values at the $\ell$'th layer, but elsewhere, superscripts for example on $\hat{\mathbf{A}}^{\ell}$ refer to powers of $\hat{\mathbf{A}}$, so that $\hat{\mathbf{A}}^{\ell} = \hat{\mathbf{A}}\cdots\hat{\mathbf{A}}$.

For objective Eq. (47) to make sense, we must assume that the normalized adjacency matrix, $\hat{\mathbf{A}}$, is invertible. In practice, $\hat{\mathbf{A}}$ may not be invertible, but we can resolve this by interpolating it with the identity,

$$\hat{\mathbf{A}}_\lambda = \lambda\mathbf{I} + (1-\lambda)\hat{\mathbf{A}}, \; \lambda \in (0, 1). \tag{48}$$

With this adjustment to the adjacency matrix, we can simplify the objective:

$$\mathcal{L}(\mathbf{G}^1, \ldots, \mathbf{G}^L) = \text{const} - \frac{\nu}{2}\sum_{\ell=1}^{L+1}\mathrm{Tr}((\hat{\mathbf{A}}\mathbf{G}^{\ell-1}\hat{\mathbf{A}}^{T})^{-1}\mathbf{G}^{\ell}). \tag{49}$$

From here, we drop the transposes because $\hat{\mathbf{A}}$ and $\mathbf{G}^{\ell}$ are symmetric. We take gradients:

$$\frac{\partial\mathcal{L}(\mathbf{G}^1, \ldots, \mathbf{G}^L)}{\partial\mathbf{G}^{\ell}} = -\frac{\nu}{2}\frac{\partial}{\partial\mathbf{G}^{\ell}}\left[\mathrm{Tr}((\hat{\mathbf{A}}\mathbf{G}^{\ell-1}\hat{\mathbf{A}})^{-1}\mathbf{G}^{\ell} + \mathrm{Tr}((\hat{\mathbf{A}}\mathbf{G}^{\ell}\hat{\mathbf{A}})^{-1}\mathbf{G}^{\ell+1})\right] \tag{50a}$$

$$= \frac{\nu}{2}\left[-(\hat{\mathbf{A}}\mathbf{G}^{\ell-1}\hat{\mathbf{A}})^{-T} + ((\mathbf{G}^{\ell})^{-1}\hat{\mathbf{A}}^{-1}\mathbf{G}^{\ell+1}\hat{\mathbf{A}}^{-1}(\mathbf{G}^{\ell})^{-1})^{T}\right]. \tag{50b}$$

Then set those gradients to zero:

$$\implies \mathbf{G}^\ell \hat{\mathbf{A}}^{-1} (\mathbf{G}^{\ell-1})^{-1} \hat{\mathbf{A}}^{-1} = \hat{\mathbf{A}}^{-1} \mathbf{G}^{\ell+1} \hat{\mathbf{A}}^{-1} (\mathbf{G}^\ell)^{-1}. \tag{51}$$

For convenience, define, $\mathbf{Z}^\ell = \mathbf{G}^\ell \hat{\mathbf{A}}^{-1} (\mathbf{G}^{\ell-1})^{-1}$, then,

$$\text{Eq. 51} \implies \mathbf{Z}^2 = \hat{\mathbf{A}} \mathbf{Z}^1 \hat{\mathbf{A}}^{-1} \tag{52a}$$

$$\mathbf{Z}^3 = \hat{\mathbf{A}} \mathbf{Z}^2 \hat{\mathbf{A}}^{-1} = \hat{\mathbf{A}}^2 \mathbf{Z}^1 \hat{\mathbf{A}}^{-2} \tag{52b}$$

$$\vdots \tag{52c}$$

$$\mathbf{Z}^\ell = \hat{\mathbf{A}}^{\ell-1} \mathbf{Z}^1 \hat{\mathbf{A}}^{-\ell+1}. \tag{52d}$$

By the definition of $\mathbf{Z}^\ell$, we have that,

$$\prod_{\ell'=1}^{\ell} \mathbf{Z}^{\ell'} = (\mathbf{G}^\ell \hat{\mathbf{A}}^{-1} (\mathbf{G}^{\ell-1})^{-1})(\mathbf{G}^{\ell-1} \hat{\mathbf{A}}^{-1} (\mathbf{G}^{\ell-2})^{-1}) \cdots ((\mathbf{G}^2 \hat{\mathbf{A}}^{-1} (\mathbf{G}^1)^{-1})(\mathbf{G}^1 \hat{\mathbf{A}}^{-1} (\mathbf{G}^0)^{-1}) \tag{53a}$$

$$= \mathbf{G}^\ell \hat{\mathbf{A}}^{-\ell} (\mathbf{G}^0)^{-1}. \tag{53b}$$

Also, by Eq. (52d), we have

$$\prod_{\ell'=1}^{\ell} \mathbf{Z}^{\ell'} = (\hat{\mathbf{A}}^{\ell-1} \mathbf{Z}^1 \hat{\mathbf{A}}^{-\ell+1})(\hat{\mathbf{A}}^{\ell-2} \mathbf{Z}^1 \hat{\mathbf{A}}^{-\ell+2}) \cdots (\hat{\mathbf{A}}^1 \mathbf{Z}^1 \hat{\mathbf{A}}^{-1})(\hat{\mathbf{A}}^0 \mathbf{Z}^1 \hat{\mathbf{A}}^0) \tag{54a}$$

$$= \hat{\mathbf{A}}^{\ell-1} (\mathbf{Z}^1 \hat{\mathbf{A}}^{-1})^\ell \hat{\mathbf{A}}. \tag{54b}$$

By setting $\ell = L + 1$, and combining equations 53b and 54b, we can solve for $\mathbf{Z}^1$,

$$\mathbf{G}^{L+1} \hat{\mathbf{A}}^{-(L+1)} (\mathbf{G}^0)^{-1} = \hat{\mathbf{A}}^L (\mathbf{Z}^1 \hat{\mathbf{A}}^{-1})^{L+1} \hat{\mathbf{A}} \tag{55a}$$

$$\implies \mathbf{Z}^1 = (\hat{\mathbf{A}}^{-L} \mathbf{G}^{L+1} \hat{\mathbf{A}}^{-(L+1)} (\mathbf{G}^0)^{-1} \hat{\mathbf{A}}^{-1})^{1/(L+1)} \hat{\mathbf{A}}. \tag{55b}$$

Matrix roots are not unique, but we calculate roots by performing an eigendecomposition. Finally we use this expression for $\mathbf{Z}^1$ to obtain a relationship between the kernel representation at layer $\ell$, $\mathbf{G}^\ell$, and the input and label kernels, $\mathbf{G}^0$ and $\mathbf{G}^{L+1}$,

$$\mathbf{G}^\ell = \hat{\mathbf{A}}^{\ell-1} ((\hat{\mathbf{A}}^{-L} \mathbf{G}^{L+1} \hat{\mathbf{A}}^{-L})(\hat{\mathbf{A}} \mathbf{G}^0 \hat{\mathbf{A}})^{-1})^{\ell/(L+1)} (\hat{\mathbf{A}} \mathbf{G}^0 \hat{\mathbf{A}}) \hat{\mathbf{A}}^{\ell-1}. \tag{56}$$

## D.1 Validity of the Solution

Note that the solution given by Eq. (56) is only valid if $\mathbf{G}_\ell$ is symmetric and positive definite. The latter property, positive definiteness, can be shown by the fact that $\mathbf{G}_\ell$ is a product of positive definite matrices. We show symmetricity with the following argument.

Let $\mathbf{C} = (\hat{\mathbf{A}}^{-L} \mathbf{G}^{L+1} \hat{\mathbf{A}}^{-L})(\hat{\mathbf{A}} \mathbf{G}^0 \hat{\mathbf{A}})^{-1}$. Since $\mathbf{C}$ is the product of two symmetric PSD matrices, it is PSD (though not necessarily symmetric). Therefore we write $\mathbf{C}$ using its eigendecomposition, $\mathbf{C} = \mathbf{V} \mathbf{D} \mathbf{V}^{-1}$, where $\mathbf{D}$ is the diagonal matrix of the eigenvalues of $\mathbf{C}$. In particular, $\mathbf{D} \geq 0$. We write $\mathbf{G}^{L+1}$ in terms of $\mathbf{V}$ and $\mathbf{D}$, and use the fact that $\mathbf{G}^{L+1}$ is a kernel matrix,

$$\mathbf{G}^{L+1} = \hat{\mathbf{A}}^L \mathbf{V} \mathbf{D} \mathbf{V}^{-1} (\hat{\mathbf{A}} \mathbf{G}^0 \hat{\mathbf{A}}) \hat{\mathbf{A}}^L \tag{57a}$$

$$= \hat{\mathbf{A}}^L (\hat{\mathbf{A}} \mathbf{G}^0 \hat{\mathbf{A}}) \mathbf{V}^{-T} \mathbf{D} \mathbf{V}^T \hat{\mathbf{A}}^L = (\mathbf{G}^{L+1})^T \tag{57b}$$

$$\implies \mathbf{V} \mathbf{D} \mathbf{V}^{-1} = (\hat{\mathbf{A}} \mathbf{G}^0 \hat{\mathbf{A}}) \mathbf{V}^{-T} \mathbf{D} \mathbf{V}^T (\hat{\mathbf{A}} \mathbf{G}^0 \hat{\mathbf{A}})^{-1}. \tag{57c}$$

This allows us to show that the powers of $\mathbf{C}$ have the following property,

$$\implies \mathbf{C}^n = (\mathbf{V} \mathbf{D} \mathbf{V}^{-1})^n = \mathbf{V} \mathbf{D}^n \mathbf{V}^{-1} = (\hat{\mathbf{A}} \mathbf{G}^0 \hat{\mathbf{A}}) \mathbf{V}^{-T} \mathbf{D}^n \mathbf{V}^T (\hat{\mathbf{A}} \mathbf{G}^0 \hat{\mathbf{A}})^{-1}. \tag{58}$$

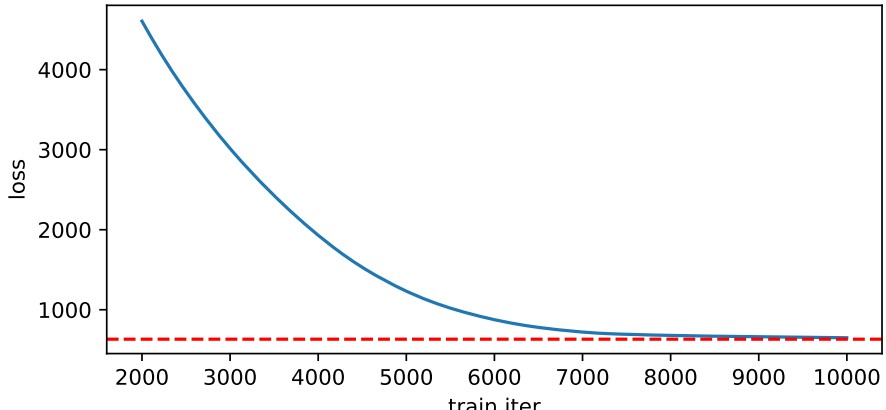

Figure 5: Losses during SGD training of the linear DKM. The loss of the analytic solution is shown by the dashed red line.

In particular, we have

$$\mathbf{C}^{\ell/(L+1)} = (\hat{\mathbf{A}}\mathbf{G}^0\hat{\mathbf{A}})\mathbf{V}^{-T}\mathbf{D}^{\ell/(L+1)}\mathbf{V}^T(\hat{\mathbf{A}}\mathbf{G}^0\hat{\mathbf{A}})^{-1}. \tag{59}$$

Therefore,

$$\mathbf{G}^\ell = \hat{\mathbf{A}}^{\ell-1}\mathbf{C}^{\ell/(L+1)}\hat{\mathbf{A}}\mathbf{G}^0\hat{\mathbf{A}}^\ell \tag{60}$$

$$= \hat{\mathbf{A}}^{\ell-1}\mathbf{V}\mathbf{D}^{\ell/(L+1)}\mathbf{V}^{-1}(\hat{\mathbf{A}}\mathbf{G}^0\hat{\mathbf{A}})\hat{\mathbf{A}}^{\ell-1} \tag{61}$$

$$= \hat{\mathbf{A}}^{\ell-1}(\hat{\mathbf{A}}\mathbf{G}^0\hat{\mathbf{A}})\mathbf{V}^{-T}\mathbf{D}^{\ell/(L+1)}\mathbf{V}^T\hat{\mathbf{A}}^{\ell-1} \tag{62}$$

$$= (\mathbf{G}^\ell)^T \tag{63}$$

as desired.

### D.2  Experimental Confirmation of Linear Analytic Results

Figures 1 and 5 demonstrate an exact match between our closed-form solution (Eq. 56) and when optimizing with gradient descent. We selected a random 200 node subset of Cora and (a) calculated the Gram matrices in closed-form, and (b) optimized a linear 2-layer graph convolutional DKM with Adam for 10000 epochs. We used a polynomial learning rate schedule, initialized at 0.1 and decaying with power 0.7.

Figure 1 visualizes the Gram matrix representations themselves, and we can see an exact match between the closed-form and gradient descent solutions. Figure 5 shows that the training loss of the gradient descent solution approaches the loss of the analytic solution (that is, the value of the linear graph convolutional DKM objective when plugging in the analytic solution).

## E  Experimental Details

For final performance metrics, we benchmarked the graph convolutional DKM on 14 node classification datasets and 5 graph classification datasets. We also benchmarked a graph convolutional NNGP and GCN on the same datasets to obtain a suitable baselines. We describe details of the datasets and the experimental method below.

### E.1  Dataset Descriptions

Here, we describe the datasets used in experiments qualitatively. The majority of the datasets we used for benchmarking were node classification datasets, where the task is to predict the labels of all nodes in a single

graph, given labels for only a limited number of nodes. Cora, Citeseer, and Pubmed (Planetoid datasets) are all citation networks, where nodes represent scientific publications, and edges are citations between those publications; one-hot keyword encodings are provided as features, and the goal is to place documents into predefined categories. Chameleon and Squirrel are both Wikipedia networks on their respective topics, so that nodes correspond to articles, and edges correspond to links between articles; features are again one-hot as in the citation networks, but represent the presence of relevant nouns, and the task is to categories pages based on their average daily traffic. Penn94 (Lim et al., 2021) is an online social network where nodes represent users, and edges represent social connections. The task is to predict the reported gender of the users. Flickr is a social network where nodes represent users, and edges correspond to friendships among users. The labels represent the interest groups of the users. Blog Catalog is a network dataset with social relationships of bloggers from the BlogCatalog website, where nodes' attributes are constructed by the keywords of user profiles. The labels represent the topic categories provided by the authors. Several of the remaining datasets were introduced by Platonov et al. (2023):

- Roman Empire is derived entirely from the Roman Empire article on Wikipedia. Nodes are words in the text, and the words are 'connected' if they are either adjacent in a sentence or if they are dependent on each other syntactically. Features are word embeddings, and the task is to label each word's syntactic role.

- Tolokers is a dataset of workers from the Toloka platform. The nodes are workers, and nodes are connected if workers have worked on the same task. Features are statistics about each worker, and the task is to predict if a worker has been banned.

- Minesweeper is a synthetic dataset. The graph has 10,000 nodes corresponding to the cells in the 100 by 100 grid in the Minesweeper game. Nodes are connected if the associated cells are adjacent. Features are one-hot encoded numbers of neighbouring mines, plus a extra feature for when the number of neighbouring mines is unknown. The task is to predict which nodes are mines.

- Amazon Ratings is a graph where nodes are Amazon products, and nodes are connected if they are commonly purchased together. Features are text embeddings of the product descriptions. The task is to predict the average user rating.

The two remaining node classification datasets, Reddit and Arxiv, are the largest (roughly one order of magnitude larger than the next biggest). In the Reddit dataset (Hamilton et al., 2017), the nodes are posts, the features are GloVe embeddings of the posts, the labels represent subreddits, and posts are connected if a user comments on both posts.

The Arxiv dataset (Hu et al., 2020b) is another citation network; nodes are computer science Arxiv papers, and papers are connected if one cites the other. Node features are Word2Vec embeddings of a paper's title and abstract, and the labels are subject areas.

We also perform experiments with graph classification datasets. The graph classification datasets all contain molecules. NCI1 and NCI109 both contain graphs of chemical compounds (nodes representing atoms, with nodes connected if there are chemical bonds between them), where the goal is to predict the effect of the compounds versus lung and ovarian cancer. Proteins is a dataset of proteins molecules, where the task is to classify as enzymes or non-enzymes. Mutag is a dataset of compounds, and the task is to predict mutagenicity on Salmonella typhimurium. Finally, Mutagenicity is a dataset of drug compounds, and the task is to categorize into mutagen or non-mutagen.

## E.2 Details of Training, Hyperparameter Selection and Final Benchmarking

### E.2.1 Node classification datasets

For node classification datasets, we used train/validation/test splits from the `torch_geometric` library (Fey & Lenssen, 2019) for most datasets; the exceptions were Arxiv and Reddit for which we adapted code from Niu et al. (2023). For data preprocessing, we (a) normalized all adjacency matrices as in Kipf &

Welling (2017), and (b) we scaled input features such that the associated linear kernel had entries in $[-1, 1]$, i.e.

$$X'_{i\lambda} = \frac{X_{i\lambda}}{\sum_\mu X_{i\mu} X_{i\mu}}. \tag{64}$$

We found that this transformation helped avoid numerical errors when training the models, as it ensures $\mathbf{G}^0 = \mathbf{X}'(\mathbf{X}')^T$ has elements all on roughly the same scale.

To select graph DKM hyperparameters, we started with a base model analogous to the GCN described by Kipf & Welling (2017), but with an extra Gram layer before the first kernel non-linearity. From this base model, we performed 4 sweeps to select (1) $\nu$ and inducing point scheme, (2) architecture, (3) centering parameters, and finally (4) the number of inducing points. The sweeps were,

- Sweep (1): $\nu \in \{0, 0.01, 0.1, 1, 10, 100, 1000, \infty\}$, `scheme` $\in \{\text{inter, intra}\}$,

- Sweep (2): `arch` $\in \{\texttt{kipf}, \texttt{kipf} + \hat{\mathbf{A}}_\lambda, \texttt{kipf} + \hat{\mathbf{A}}_\lambda + \texttt{resid}, \texttt{platinov}\}$. Here, `kipf` refers to the default architecture, $+\hat{\mathbf{A}}_\lambda$ means that $\hat{\mathbf{A}}_\lambda$ was used instead of the usual renormalized adjacency matrix, $+\texttt{resid}$ refers to the addition of residual blocks, and `platinov` refers to an architecture similar to the one used in Platonov et al. (2023). The `platinov` architecture is similar to a residual 2-layer GCN, but with two linear layers in each residual block.

- Sweep (3) searched over centering or no centering, and whether to include a learned scale and bias parameter (described in Appendix F).

- Sweep (4): $P_i \in \{50, 100, 200, 300, 400\}$.

We trained for 300/200 epochs in sweep (1), and in the remaining sweeps we trained for 200/150 for the smaller/bigger datasets (where Arxiv and Reddit and the 'bigger' datasets). For the smaller datasets 'one epoch' means a single full-batch gradient descent step; for the bigger datasets it means two mini-batched gradient descent steps.

We had a numerical stability issue with Squirrel when selecting hyperparameters. To resolve this, we set `arch=kipf`, rather than using `arch=platinov`, which was the 'best' according to sweep (2).

We repeated the above sweeps, but with DKM regularization fixed to $\nu = \infty$ (equivalent to removing Gram layers) to obtain final performance metrics for a sparse graph convolutional NNGP. To find suitable hyperparameters for the GCN, we used a single grid search starting with the base GCN model of Kipf & Welling (2017). We searched over `width` $\in \{100, 200\}$, `dropout` $\in \{0, 0.5\}$, `batchnorm` $\in \{\text{yes, no}\}$, and `row_normalization` $\in \{\text{yes, no}\}$, `arch` $\in \{\texttt{kipf}, \texttt{platinov}\}$.

We used the Adam optimizer with a two-stage learning rate schedule for all training runs. We increase the learning rate linearly from $10^{-3}$ to $10^{-2}$ for the first quarter of the epochs, and after that use a cosine schedule with a minimum learning rate of $10^{-5}$. The models were written in Pytorch, and we trained on a cluster containing RTX 2080's, RTX 3090's and A100s.

### E.2.2  Graph classification datasets

For graph classification datasets, we constructed our own cross-validation splits (10 splits), since no default split was provided by `torch_geometric`. We adopted a very similar approach for hyperparameter selection as in Section E.2.1. The main differences being that we set `scheme`=inter (since an intra-domain inducing-point scheme is not applicable in the multi-graph setting), and we trained for 300 batches on all datasets.

To perform graph classification with node-level features/kernels of multiple graphs, we (a) stack graphs together in batches of 1024, and (b) perform a mean aggregation over nodes in each graph at the top layer of the network, thus obtaining logits for each graph.

### E.3  Dataset Statistics

We include dataset statistics for node classification datasets in Table 3 and Table 4 respectively.

Table 3: Node classification dataset statistics.

| Dataset | # nodes | # edges | Homophily ratio | # features | # classes |
|---|---|---|---|---|---|
| Roman Empire | 22,662 | 32,927 | 0.05 | 300 | 18 |
| Squirrel | 5,201 | 198,353 | 0.22 | 2,089 | 5 |
| Flickr | 7,575 | 239,738 | 0.24 | 12,047 | 9 |
| Chameleon | 2,227 | 31,371 | 0.23 | 2,325 | 5 |
| Amazon Ratings | 24,492 | 93,050 | 0.38 | 300 | 5 |
| Blog Catalog | 5,196 | 171,743 | 0.40 | 8,189 | 6 |
| Penn94 | 41,554 | 1,362,229 | 0.47 | 4,814 | 2 |
| Tolokers | 11,758 | 519,000 | 0.59 | 10 | 2 |
| Arxiv | 169,343 | 1,157,799 | 0.65 | 128 | 40 |
| Minesweeper | 10,000 | 39,402 | 0.68 | 7 | 2 |
| Citeseer | 3,327 | 4,552 | 0.74 | 3,703 | 6 |
| Reddit | 232,965 | 57,307,946 | 0.76 | 602 | 41 |
| Pubmed | 19,717 | 44,324 | 0.80 | 500 | 3 |
| Cora | 2,708 | 5,278 | 0.81 | 1,433 | 7 |

Table 4: Graph classification dataset statistics.

| Dataset | # graphs | avg. # nodes | avg. # edges | # features | # classes |
|---|---|---|---|---|---|
| Proteins | 1,113 | 39.1 | 72.8 | 5 | 2 |
| NCI1 | 4,110 | 29.9 | 32.3 | 38 | 2 |
| NCI109 | 4,127 | 29.7 | 32.1 | 39 | 2 |
| Mutag | 188 | 17.9 | 19.8 | 8 | 2 |
| Mutagenicity | 4,337 | 30.3 | 30.8 | 15 | 2 |

### E.4 Details of the Other Experiments

For Figure 1 and Figure 5, we trained a linear graph convolutional DKM for 10,000 epochs on a randomly selected 200 node subset of Cora.

For Figure 2, we calculate the linear graph convolutional DKM kernels on randomly generated Erdős-Rényi graphs, with 50 nodes and edge probability 0.1. When calculating the kernels, we use $\hat{\mathbf{A}}_{\lambda=0.5}$ for the graph mixup. The input kernels are standard Wishart samples with 50 degrees of freedom. The nodes are randomly assigned one of two classes, with an even split.

For Figure 3 we trained graph convolutional DKMs with the arc-cosine kernel non-linearity for 300 epochs, with regularization $\nu \in \{0, 1, 10, 1000\}$, and $\hat{\mathbf{A}}_{\lambda=0.3}$ for graph mixup. The kernel alignment (CKA) figures were computed using the full kernels, without any normalization.

For all of the Figures described above in this subsection, we used a simple 2-layer architecture, similar to Kipf & Welling (2017). All kernel plots are normalized to ensure entries are between $[-1, 1]$.

Finally, to produce Figure 4 we used data from the first hyperparameter sweep.

## F  Kernel Centering Layer

As part of our grid search, we considered adding a 'kernel centering layer'. When enabled, the centering layer performs a centering similar to batchnorm,

$$(\mathbf{F}_t)'_{i\lambda} = (\mathbf{F}_t)_{i\lambda} - \frac{1}{P_t} \sum_j (\mathbf{F}_t)_{j\lambda}, \tag{65}$$

where $\mathbf{K} = \mathbf{F}_t \mathbf{F}_t^T$ is the kernel to be centered. Additionally, we include the option of learning a scale parameter $\gamma$ and bias parameter $\beta$,

$$(\mathbf{F}_t)''_{i\lambda} = \gamma(\mathbf{F}_t)'_{i\lambda} + \beta. \tag{66}$$

## G  Inducing-point ablations

Table 5 shows validation accuracies for different inducing point schemes. We obtained these accuracies during Sweep (1), so that $\nu$ has been tuned for each inducing-point scheme (but not other hyperparameters). We find that inter-domain is a better scheme overall, but intra-domain can be beneficial for a few datasets (most notably Penn94).

Table 5: Validation accuracies for the two inducing point scheme on a range of node classification datasets.

|  | Inter-domain | Intra-domain |
| --- | --- | --- |
| Cora | **78.9 ± 0.2** | 58.2 ± 0.5 |
| Pubmed | **79.4 ± 0.4** | 74.4 ± 1.6 |
| Reddit | 94.2 ± 0.0 | **94.5 ± 0.0** |
| Citeseer | **70.8 ± 0.2** | 23.4 ± 0.2 |
| Minesweeper | **80.2 ± 0.2** | 80.1 ± 0.3 |
| Arxiv | 69.6 ± 0.1 | **70.2 ± 0.1** |
| Tolokers | 79.9 ± 0.4 | **80.5 ± 0.5** |
| Penn94 | 69.5 ± 0.3 | **81.3 ± 0.4** |
| BlogCatalog | **76.0 ± 0.8** | 72.8 ± 1.3 |
| Amazon Ratings | 46.3 ± 0.3 | **47.8 ± 0.5** |
| Flickr | **59.2 ± 1.6** | 55.8 ± 1.6 |
| Chameleon | **66.5 ± 2.0** | 66.0 ± 2.0 |
| Squirrel | 49.9 ± 0.7 | **54.4 ± 0.9** |
| Roman Empire | 51.2 ± 0.7 | **51.7 ± 0.8** |

## H  Deep Graph Convolutional DKMs

We investigated how the graph convolutional DKM performs as we increase the number of layers, with results shown in Figure 6. Similarly to the inducing point ablations (Section G), we show validation accuracies using settings found in Sweep (1), so that $\nu$ and inducing-point scheme have been tuned (but not other hyperparameters). We find that increasing depth tends not to help performance (though it can occassionally help in some cases, e.g. Citeseer). This finding is expected, since GCN also loses performance as depth is scaled (Oono & Suzuki, 2019). Note that we encountered a numerical instability when running the Reddit depth experiment, so the relevant data points have been omitted (depth=6 and 8).

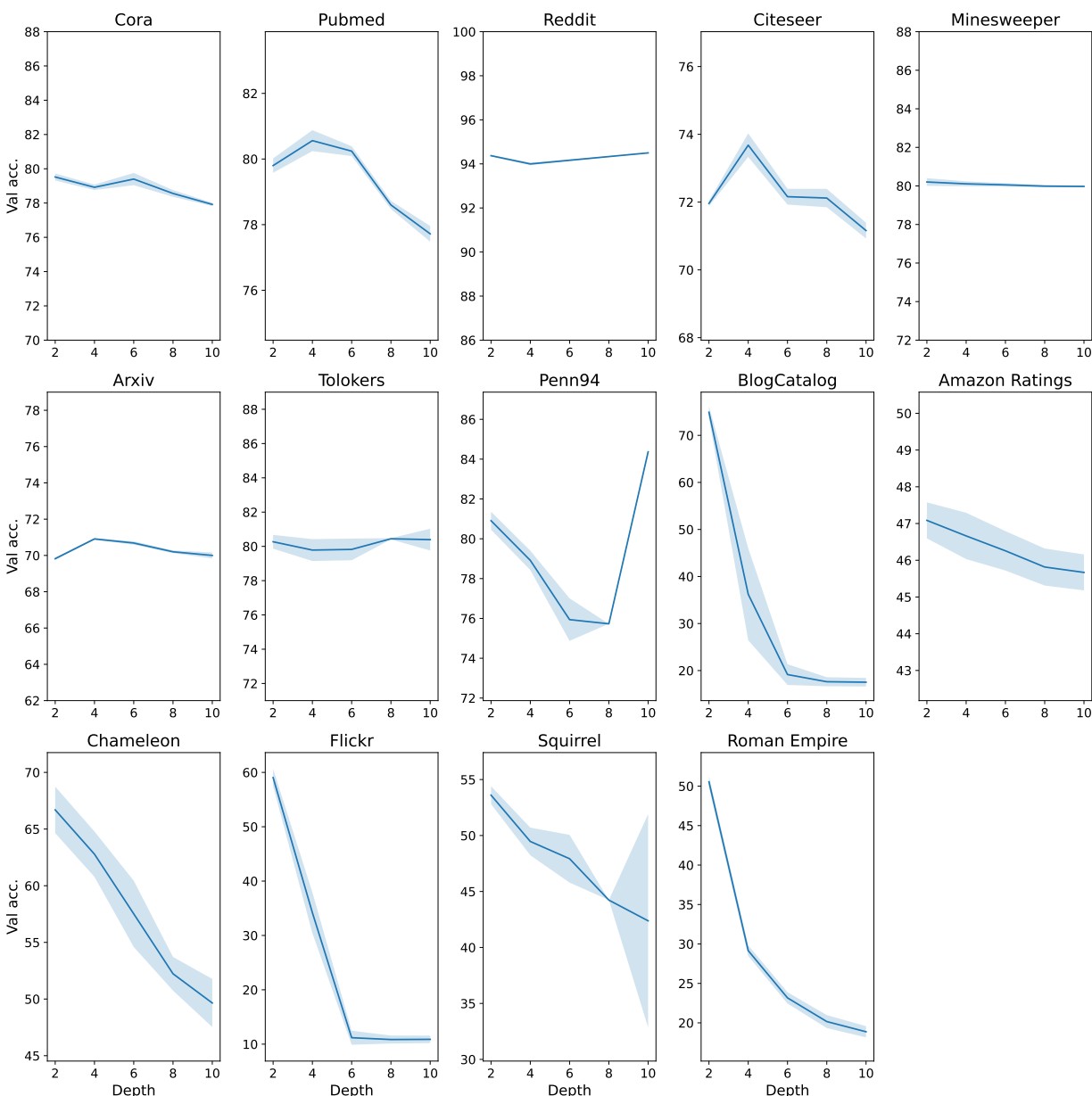

Figure 6: Validation accuracy at different depths for different node classification datasets. Error bands are $\pm$ 1 standard deviation.

# I   Parameterization of the Deep Kernel Machine

Even though the inducing point scheme significantly reduces the computational complexity of training a DKM, computing the objective and forward equations is still expensive and has the potential to be unstable. In particular, the KL regularization terms

$$D_{\mathrm{KL}}\big(\mathcal{N}(\mathbf{0},\mathbf{G}^{\ell}) \,||\, \mathcal{N}(\mathbf{0},\mathbf{K}(\mathbf{G}^{\ell-1}))\big) =$$
$$\frac{1}{2}\big(P - \log\det(\mathbf{K}(\mathbf{G}^{\ell-1})^{-1}\mathbf{G}^{\ell}) + \mathrm{Tr}(\mathbf{K}(\mathbf{G}^{\ell-1})^{-1}\mathbf{G}^{\ell})\big) \tag{67}$$

involve an unpleasant inverse and log-determinant. When implementing convolutional DKMs, Milsom et al. (2023) performed these calculations naively, which necessitates the use of double precision to avoid stability issues. The use of double precision is inefficient.

We propose the following parameterization to avoid the $\log \det$, while still computing the regularization term exactly. At each layer, we learn the parameter $\mathbf{L}^\ell = \text{cholesky}(\mathbf{K}(\mathbf{G}_{\text{ii}}^{\ell-1})^{-1}\mathbf{G}_{\text{ii}}^\ell)$. Due to the fact that $\mathbf{L}^\ell$ is lower triangular, the log determinant term can easily be computed as,

$$\log \det(\mathbf{K}(\mathbf{G}_{\text{ii}}^{\ell-1})^{-1}\mathbf{G}_{\text{ii}}^\ell) = \log \det(\mathbf{L}\mathbf{L}^T) = 2\sum_{j=1}^{P_{\text{i}}} \log L_{jj}^\ell. \tag{68}$$

The forward equations given by Yang et al. (2023) are

$$\mathbf{G}_{\text{ti}}^\ell = \mathbf{K}_{\text{ti}}\mathbf{K}_{\text{ii}}^{-1}\mathbf{G}_{\text{ii}}^\ell, \tag{69a}$$

$$\mathbf{G}_{\text{tt}}^\ell = \mathbf{K}_{\text{tt}} - \mathbf{K}_{\text{ti}}\mathbf{K}_{\text{ii}}^{-1}\mathbf{K}_{\text{it}} + \mathbf{K}_{\text{ti}}\mathbf{K}_{\text{ii}}^{-1}\mathbf{G}_{\text{ii}}^\ell\mathbf{K}_{\text{ii}}^{-1}\mathbf{K}_{\text{it}}. \tag{69b}$$

We suggest using the Nyström approximation $\mathbf{K}_{\text{tt}} \approx \mathbf{K}_{\text{ti}}\mathbf{K}_{\text{ii}}^{-1}\mathbf{K}_{\text{it}}$ to avoid computing $\mathbf{K}_{\text{ti}}\mathbf{K}_{\text{ii}}^{-1}\mathbf{K}_{\text{it}}$. Additionally, we know that there exist $\mathbf{F}_{\text{i}}$ and $\mathbf{F}_{\text{t}}$ such that

$$\begin{pmatrix} \mathbf{G}_{\text{ii}}^\ell & \mathbf{G}_{\text{it}}^\ell \\ \mathbf{G}_{\text{ti}}^\ell & \mathbf{G}_{\text{tt}}^\ell \end{pmatrix} = \begin{pmatrix} \mathbf{F}_{\text{i}}\mathbf{F}_{\text{i}}^T & \mathbf{F}_{\text{i}}\mathbf{F}_{\text{t}}^T \\ \mathbf{F}_{\text{t}}\mathbf{F}_{\text{i}}^T & \mathbf{F}_{\text{t}}\mathbf{F}_{\text{t}}^T \end{pmatrix}. \tag{70}$$

Therefore the Gram forward equations (Eq. 69) with a Nystrom approximation for $\mathbf{K}_{\text{tt}}$ are equivalent to

$$\mathbf{H}_{\text{i}} = \text{cholesky}(\mathbf{K}_{\text{ii}}), \tag{71a}$$

$$\mathbf{F}_{\text{i}} = \mathbf{H}_{\text{i}}\mathbf{L}^\ell, \tag{71b}$$

$$\mathbf{F}_{\text{t}} = \mathbf{K}_{\text{ti}}\mathbf{H}_{\text{i}}^{-1}\mathbf{L}^\ell. \tag{71c}$$

Notably, the $\mathbf{K}_{\text{ti}}\mathbf{H}_{\text{i}}^{-1}$ computation can be achieved using a triangular solve, since $\mathbf{H}_{\text{i}}$ is lower triangular. The trade-off of propagating with Eq. (71) rather than the original scheme (Eq. 69) is that in exchange for using the Nyström approximation we get (a) efficient objective computation avoiding $\log \det$, (b) a triangular solve instead of a cholesky solve, and (c) fewer matrix multiplies.

