# OpenReview forum: "Flexible Infinite-Width Graph Convolutional Neural Networks"
_TMLR — Accepted by TMLR_

### Review · Reviewer_Qhdi · 2024-12-22

**Summary Of Contributions:**

Development of Graph Convolutional DKMs: introduce a variant of deep kernel machines tailored for graph data, which allows for adjustable representation learning, overcoming the static nature of traditional NNGPs.

Empirical Evaluation: provide empirical evidence showing that representation learning significantly improves performance in heterophilic node classification tasks, highlighting the model's utility in scenarios where node labels are dissimilar to those of their neighbors​.

**Audience:**

Yes

**Claims And Evidence:**

Yes

**Requested Changes:**

The technical introduction from page 3-6 is too long. I suggest the authors to move the main content, especially the formulas, to appendix and summarize the main idea within two pages.

**Strengths And Weaknesses:**

## Strengths

Model Design: The introduction of graph convolutional DKMs extends neural network Gaussian processes (NNGPs) to graph convolutional networks (GCNs), which allows for tunable representation learning.

Empirical Analysis: The empirical evaluations demonstrate the efficacy of the proposed model, especially in heterophilic node classification tasks.

The related works are organized very well


## Weaknesses and Questions

1. Lack of contribution: as the authors state, equation (22) follows equation (14), the inducing point scheme follows [1]. What is the main contribution beyond the simple extension of the previous work? I hope the authors can emphasize and highlight it.
2. The modified adjacency matrix defined in equation (26) is the same as the generalized lazy random walk matrix in [2].
3. The propose method increases the complexity of hyperparameter selection.
4. "We see that more homophilous datasets like Cora tend to have stronger performance for larger ν", This seems only correct for Cora, but not for other homophilic graphs. Even for those $h>0.65$, they do not have the similar curves. For the heterophilic graphs, it looks like that their curves share a certain pattern. However, it is recently found that only considering homophily/heterophily is not enough to analyze the behavior of graph models [3,4]. We should consider both label, structure and feature aspects [4].
5. Can your study be extended from $\hat{A}$ to other graph filters or learning techniques, e.g. high-pass filters [2,3] and negative message passing [5,6]? These methods are found to be effective for heterophily problem. (This is not a weakness, just a question)
6. How does your proposed method compare with SOTA models, e.g. [3,5,6]?
7. Since graph convolutional DKM doesn't show clear edge over baseline GCN in terms of prediction accuracy, could you please clarify the biggest advantage of your proposed methods in practice, e.g. faster training or efficient inference?
8. Most of the heterophilic datasets you test on are not hard datasets [7]. I recommend to verify your model on more challenging heterophilic datasets, e.g. the malignant and ambiguous heterophilic datasets [7]. (This is not a weakness)

[1] A theory of representation learning in deep neural networks gives a deep generalisation of kernel methods. International Conference on Machine Learning, 2023. In press.

[2] Complete the Missing Half: Augmenting Aggregation Filtering with Diversification for Graph Convolutional Networks. InNeurIPS 2022 Workshop: New Frontiers in Graph Learning 2022 Nov 22.

[3] Revisiting heterophily for graph neural networks. Advances in neural information processing systems. 2022 Dec 6;35:1362-75.

[4] What is missing in homophily? disentangling graph homophily for graph neural networks[J]. arXiv preprint arXiv:2406.18854, 2024.

[5] Beyond low-frequency information in graph convolutional networks. In Proceedings of the AAAI Conference on Artificial Intelligence, volume 35, pp. 3950–3957, 2021.

[6] Adaptive universal generalized pagerank graph neural network. In International Conference on Learning Representations, 2021.

[7] The heterophilic graph learning handbook: Benchmarks, models, theoretical analysis, applications and challenges. arXiv preprint arXiv:2407.09618. 2024 Jul 12.

---

> ### Author Response · Authors · 2025-02-26
>
> Thank you for your careful and insightful review!
>
> > Lack of contribution
>
> We develop machinery necessary to apply DKMs to graphs, and use them to address theoretical and empirical questions.
>
> In particular, development of DKMs for the graph setting mirrors work introducing other methods for the graph setting, such as development of GCNs.
> GCNs are regarded as making a critical contribution, despite being heavily inspired by previous work on NNs.
> In GCNs, the technical challenge was propagating information node-to-node, and the solution was to average representations of connected nodes.
> Obviously, this simple solution is highly effective, and hugely impactful.
> But it was very simple.
> In contrast, in GCDKMs, the key technical challenge is to develop an inducing-point inference scheme that operates in the graph setting.
> Obviously, inducing-point inference schemes are by their nature, far more technically challenging than averaging representations across nodes.
> To our knowledge, no such inducing-point scheme for graph nodes exists in any literature.
> Moreover, the most obvious scheme to take inspiration from, that of convolutional DKMs, is in actual fact very different.
> For convolutional DKMs the `inducing points' do not have a clear interpretation in terms of e.g. an inducing patch, and are designed abstractly.
> In contrast, here we use an entirely new concept: that of inducing nodes.
> The novel technical challenge inherent in our scheme is embodied in the full derivation in the appendices, which are highly involved.
>
> But addressing these technical challenges was only our first contribution, we made two further contributions (both noted in the reviewer's strengths section):
>
> Our second contribution was to assess the empirical performance of GCDKMs vs previous methods.
>
> Our final contribution was to exploit the GCDKM's "knob" that tunes the degree of representation learning.
> This allows us to study the importance of representation learning in various tasks.  We found representation learning gives noticeable performance improvements for heterophilous node classification but less so for homophilous node classification.
>
> > The modified adjacency matrix...
>
> We added a reference in the revision.
>
> > The propose method increases the complexity of hyperparameter selection.
>
> Yes. Given these methods are at an early stage of development, we hope future work will improve our understanding of DKMs and reduce the additional complexity.
>
> > "We see that more homophilous datasets like Cora tend to have stronger performance for larger ν", This seems only correct for Cora...
>
> We could have been clearer here. We replaced this with 'For more homophilous datasets (see top row of Figure 4) like Cora, performance tends to remain relatively stable or even improve with larger v, suggesting that the standard architectural inductive biases are well-suited to these tasks. In contrast, heterophilous datasets such as Roman Empire benefit from smaller v (i.e.\ increased flexibility).'
>
> > Can your study be extended from to other graph filters or learning techniques...?
>
> Yes, these are exciting areas for future work.  It should be possible to "kernelize" any GCN method and thereby import it into the DKM setting.
>
> > How does your proposed method compare with SOTA models...?
>
> As discussed above, we can develop DKM extensions analogous to these methods by kernelising [3,5,6]. We believe a fair comparison is [3,5,6] vs corresponding extensions of the DKM.  Unfortunately, these are non-trivial extensions of GCNs and as such, their kernelisation is also non-trivial, and is thus out-of-scope for this manuscript. Nonetheless, we are able to speculate on the results: since [3,5,6] build in better inductive biases, they should improve the performance of all methods (GCDKM, GCNNGP and GCN). We actually think they would give most marked improvements to the GCNNGP, as representation learning in the GCDKM and GCN is able to compensate for imperfect inductive biases, whereas GCNNGPs have no representation learning, so are very sensitive to the built-in inductive biases.
>
> We have included a discussion of these points in the limitations section.
>
> > please clarify the biggest advantage of your proposed methods in practice...?
>
> GCDKMs give critical insights into the importance of representation learning for graph tasks. We found that representation learning gives noticeable performance improvements for heterophilous node classification tasks, but less so for homophilous node classification tasks.
>
> > verify your model on more challenging heterophilic datasets
>
> We extended the results with three malignant/ambiguous datasets from [7]: Penn94, Flickr, and BlogCatalog datasets. See Figure 4, and Table 1. These datasets match the qualitative patterns that we observed in the original submission.
>
> > technical introduction from page 3-6 is too long...
>
> We cut down the background section to approximately 2.5 pages, but have not been able to get it within two pages.

---

### Review · Reviewer_tTD2 · 2025-01-17

**Summary Of Contributions:**

This paper mainly studies the abilities of NNGP and graph convolutional DKM in graph node classification tasks. This paper first introduces how to formulate the NNGP and DKM based on graph convolutional network, and then performs empirical studies to evaluate their performance for various node classification tasks. This paper shows that the representation induced by DKM can be beneficial for heterophilous node classification, while the  performance improvement is marginal for homophilous node classification tasks.

The contributions are summarized as follows:

* This paper provides a clean formula for the NNGP and DKM in graph convolution networks and build a framework for performing different trade-off between NNGP and DKM.
* This paper empirically studies the benefit of representation learning in various graph tasks, and finds different patterns in heterophilous and homophilous node classification tasks.

**Audience:**

Yes

**Broader Impact Concerns:**

No concern.

**Claims And Evidence:**

Yes

**Requested Changes:**

* Add some theoretical analysis to better show the difference between DKM and NNGP for different tasks, at least for some simple tasks.
* Add more experiments regarding the graph classification problems and identify the difference between DKM and NNGP.

**Strengths And Weaknesses:**

The strengths are as follows:

* the rigorous derivations for NNGP and DKM.
* Well-organized presentation, which is easy to follow.
* A inducing-point scheme that can compute the graph convolution DKM with linear scaling in dataset size.

The weaknesses are as follows:

* This paper only provides empirical results for NNGP and DKM in graph node classification tasks. All results are straightforwardly obtained via experiments. However, it lacks a detailed discussion and explanation for the empirical observations, especially why NNGP can be good for some particular tasks while DKM is good for other tasks. This requires preciser designs of the experiments or theoretical analysis.

* This paper only studies node classification problems, while graph convolution networks are also widely used for graph classification problems. This direction should also be considered in the paper as kernel machines may perform different for different graph tasks.

---

> ### Author Response · Authors · 2025-02-26
>
> Thank you for your positive review, which emphasises:
> * the rigorous derivations for NNGP and DKM.
> * Well-organized presentation, which is easy to follow.
> * An inducing-point scheme that can compute the graph convolution DKM with linear scaling in dataset size.
>
> >This paper only provides empirical results for NNGP and DKM in graph node classification tasks. All results are straightforwardly obtained via experiments. However, it lacks a detailed discussion and explanation for the empirical observations, especially why NNGP can be good for some particular tasks while DKM is good for other tasks. This requires preciser designs of the experiments or theoretical analysis.
>
> We believe that the differences between NNGPs and DKMs are best addressed empirically, as the differences are highly dataset-dependent.
> In particular, in the limit as $\nu \rightarrow \infty$, the DKM becomes an NNGP.
> Figure 4 analyses this limit, finding that taking $\nu$ large hurts you a lot in heterophilous classification tasks, but not in homophilous.
> We believe this is basically down to the appropriateness of the inductive biases for the data.
> If the inductive biases are very well-suited to the data, then the representations are good, then both the DKM and the NNGP (or equivalently, the NNGP with large $\nu$) does well.
> This is the case for homophilous classification tasks.
> In contrast, for heterophilous classification tasks, it seems that the inductive biases from the standard architecture are less appropriate.
> The GCDKM can use representation learning to compensate for these poor inductive biases, but the GCNNGP does not have representation learning, so cannot compensate, and does poorly.
> We have extended our discussion in Section 6.2 to include these points.
>
> >This paper only studies node classification problems, while graph convolution networks are also widely used for graph classification problems. This direction should also be considered in the paper as kernel machines may perform different for different graph tasks.
>
> We added some results for graph classification tasks (please see Table 2) and an overall summary of the results in Section 6.3. The results of the GCDKM are similar to those of the GCN, especially when we take error bars into account.

---

### Review · Reviewer_5xCQ · 2025-02-10

**Summary Of Contributions:**

The paper introduces the graph convolutional deep kernel machine (DKM), a flexible variant of infinite-width graph convolutional networks that allows for representation learning through a tunable parameter v. The authors develop scalable inducing-point approximation schemes to make the model computationally feasible for large datasets and provide a closed-form solution for learned representations in the linear setting, offering theoretical insights. Empirically, they demonstrate that representation learning significantly improves performance in heterophilous node classification tasks (where connected nodes tend to have dissimilar labels) but has less impact on homophilous tasks (where connected nodes tend to have similar labels). The graph convolutional DKM performs comparably to finite-width graph convolutional networks (GCNs) on most datasets, bridging the gap between infinite-width and finite-width models. This work advances the understanding of representation learning in graph neural networks, particularly in heterophilous settings, and provides a scalable framework for applying infinite-width models to large-scale graph data.

**Audience:**

Yes

**Broader Impact Concerns:**

This work advances the theoretical and practical understanding of infinite-width neural networks in graph-based learning, introducing a flexible framework that enhances representation learning. While we do not foresee direct ethical concerns, potential considerations include biases inherited from graph structures, computational efficiency, and interpretability of learned representations. Future work could explore fairness-aware learning, scalable kernel approximations, and improved explainability to further enhance the applicability of Graph Convolutional Deep Kernel Machines.

**Claims And Evidence:**

Yes

**Requested Changes:**

- Include a comparison to other infinite-width frameworks, such as the Neural Tangent Kernel (NTK) [1] and \muP parameterized models, in the graph node classificaiton setting.

- Explore Deeper Architectures: Investigate the performance of deeper architectures (e.g., 4-layer or 6-layer models) and the impact of residual connections on representation learning.

- Ablation on Inducing-Point Schemes: Perform an ablation study to compare the two inducing-point schemes (intra- and inter-domain) in more detail, including their impact on performance and computational cost.

- Clarify Limitations: Expand the discussion of limitations, particularly regarding computational cost and scalability, and suggest potential avenues for future work to address these issues.

[1] Huang, W., Li, Y., Du, W., Yin, J., Da Xu, R.Y., Chen, L. and Zhang, M., 2021. Towards deepening graph neural networks: A GNTK-based optimization perspective. arXiv preprint arXiv:2103.03113.

**Strengths And Weaknesses:**

- Novel Framework: Introduces the graph convolutional DKM, a flexible infinite-width model that bridges the gap between fixed-kernel NNGPs and finite-width GCNs by enabling controlled representation learning via the parameter v.

- Scalability: Proposes two inducing-point approximation schemes (intra- and inter-domain) to reduce computational costs, making the model feasible for large graphs.

- Empirical Breadth: Evaluates performance across 11 datasets with varying homophily levels, demonstrating that representation learning is critical for heterophilous tasks and less so for homophilous ones.

---

> ### Author Response · Authors · 2025-02-26
>
> Thanks for your positive and insightful review, which emphasizes:
>
> > Novel Framework: Introduces the graph convolutional DKM, a flexible infinite-width model that bridges the gap between fixed-kernel NNGPs and finite-width GCNs by enabling controlled representation learning via the parameter v.
>
> > Scalability: Proposes two inducing-point approximation schemes (intra- and inter-domain) to reduce computational costs, making the model feasible for large graphs.
>
> > Empirical Breadth: Evaluates performance across 11 datasets with varying homophily levels, demonstrating that representation learning is critical for heterophilous tasks and less so for homophilous ones.
>
> In terms of comments:
>
> ### Comparison to other infinite width frameworks
>
> We already have extensive results for one infinite-width framework, the (GC)NNGP, and the reviewer acknowledges ``Empirical Breadth: Evaluates performance across 11 datasets with varying homophily levels, demonstrating that representation learning is critical for heterophilous tasks and less so for homophilous ones.''
>
> In practice, people find that NTK gives very similar performance to the NNGP in almost all settings tested[B], and there are weak indications that the NNGP approach might be slightly better than the NTK [A].
> Given that, it is of marginal benefit to add results for the NTK.
> Finally, while mu-P does use an infinite width limit, it uses that limit to achieve a quite different goal, and does not give a kernel (or, if it does give a kernel, it is simply the NNGP or NTK kernel).
> In particular, mu-P gives scaling laws relating the optimal learning rate to network width.
> These scaling laws are derived in the infinite-width limit.
> But in practice, they are used exclusively in finite-width networks.
> In particular, to use mu-P, you find the optimal learning rate in a small scale setting using many training runs.
> Then, you scale up the network width for a final, very expensive, large-scale run.
> mu-P tells you how to transfer the optimal learning rate from the small-scale setting to the final large-scale run, so you don't need to do hyperparameter optimization in the large-scale setting.
>
> ### Explore Deeper Architectures
>
> We performed an experiment that investigates the effect of depth on performance. Please see Appendix H, Figure 6 for results.
>
> ### Ablation on Inducing-Point Schemes:
>
> We have added an ablation that compares performance between the two schemes, please see Appendix G, Table 5. The difference in computational cost between the two schemes is negligible because the overall cost is dominated by other computations --- we included a comment about this in Section 4.2. This ablation shows that Inter-domain is usually always the same as or better than Intra-domain, with the exception of three datasets: Penn94, Flickr and Squirrel.
>
> ### Clarify Limitations
>
> We have added a limitations section.  The key limitation of GCDKMs, like kernel methods in general, is scaling to large dataset sizes due to the cubic cost of naive methods.
> We implemented an in inducing point scheme to circumvent this limitation, but it may be possible to improve on these methods.
>
> [A]: Lee, Jaehoon, et al. "Finite versus infinite neural networks: an empirical study." (2020)
>
> [B]: Li, Zhiyuan, et al. "Enhanced convolutional neural tangent kernels." (2019).

---

### Author Response · Authors · 2025-02-26

We would like to thank the reviewers for their positive reviews, and interesting and insightful comments, that have improved the paper.

The reviewers unanimously affirmed that the paper meets the two criteria for publication in TMLR:
* Are the claims made in the submission supported by accurate, convincing and clear evidence?
* Would some individuals in TMLR's audience be interested in the findings of this paper?

To summarise the paper, we don't think we can do better than 5xCQ:

> The paper introduces the graph convolutional deep kernel machine (DKM), a flexible variant of infinite-width graph convolutional networks that allows for representation learning through a tunable parameter v. The authors develop scalable inducing-point approximation schemes to make the model computationally feasible for large datasets and provide a closed-form solution for learned representations in the linear setting, offering theoretical insights. Empirically, they demonstrate that representation learning significantly improves performance in heterophilous node classification tasks (where connected nodes tend to have dissimilar labels) but has less impact on homophilous tasks (where connected nodes tend to have similar labels). The graph convolutional DKM performs comparably to finite-width graph convolutional networks (GCNs) on most datasets, bridging the gap between infinite-width and finite-width models. This work advances the understanding of representation learning in graph neural networks, particularly in heterophilous settings, and provides a scalable framework for applying infinite-width models to large-scale graph data.

Further, the reviewers agreed on:

The novelty and rigour of the model design:

> Novel Framework: Introduces the graph convolutional DKM, a flexible infinite-width model that bridges the gap between fixed-kernel NNGPs and finite-width GCNs by enabling controlled representation learning via the parameter v. (5xCQ)

> the rigorous derivations for NNGP and DKM. (tTD2)

The breadth of the experiments:

> Empirical Breadth: Evaluates performance across 11 datasets with varying homophily levels, demonstrating that representation learning is critical for heterophilous tasks and less so for homophilous ones. (5xCQ)

> Empirical Analysis: The empirical evaluations demonstrate the efficacy of the proposed model, especially in heterophilic node classification tasks. (Qhdi)

The clarity of the presentation:

> Well-organized presentation, which is easy to follow. (tTD2)

> The related works are organized very well (Qhdi)

And the scalability of the inference scheme (not a given for kernel methods):

> Scalability: Proposes two inducing-point approximation schemes (intra- and inter-domain) to reduce computational costs, making the model feasible for large graphs. (5xCQ)

---

### Decision · Action_Editor_5S9s · 2025-05-14

**Recommendation:** Accept as is

**Comment:**

This paper introduces the graph convolutional deep kernel machine (DKM), an infinite-width model inspired by NNDP and GCN, to enhance representation learning. The authors develop scalable approximation schemes to make the model feasible for large datasets, and offer theoretical insights and closed-form solutions in linear settings. Empirical results show that representation learning significantly enhances performance in heterophilous node classification tasks, while having less impact on homophilous tasks, thereby advancing the understanding of representation learning in graph neural networks. While reviewers noted that the paper's novelty does not stand out significantly, they also agree that the proposed methods are solid and interesting.

**Audience:**

This paper is of a general interest to researchers on graph convolutional neural networks and learning theory related to neural network Gaussian process and neural tangent kernel.

**Claims And Evidence:**

The paper presents convincing methods and results. The reviewers all agree that the proposed graph convolutional deep kernel machine is rigorous and the inducing-point scheme that achieves linear scaling in dataset size is interesting.